# A Tumor-Specific Molecular Network Promotes Tumor Growth in *Drosophila* by Enforcing a Jun N-Terminal Kinase–Yorkie Feedforward Loop

**DOI:** 10.3390/cancers16091768

**Published:** 2024-05-02

**Authors:** Indrayani Waghmare, Karishma Gangwani, Arushi Rai, Amit Singh, Madhuri Kango-Singh

**Affiliations:** 1Department of Biology, University of Dayton, Dayton, OH 45469, USA; indrayani_waghmare@uml.edu (I.W.); raia05@udayton.edu (A.R.); asingh1@udayton.edu (A.S.); 2Department of Biological Sciences, University of Massachusetts Lowell, Lowell, MA 01854, USA; 3Computational Biology Department, St Jude Children’s Research Hospital, Memphis, TN 38105, USA; 4Premedical Programs, University of Dayton, Dayton, OH 45469, USA; 5Integrative Science and Engineering Centre (ISE), University of Dayton, Dayton, OH 45469, USA

**Keywords:** apical basal polarity, cancer, *Drosophila*, imaginal discs, Wingless, Hippo, JNK, caspase

## Abstract

**Simple Summary:**

Cancer genomics and transcriptomics have revealed genes and pathways altered in several cancers. These studies have provided valuable information about cancer cells, their origin, oncogenic processes, and signaling pathways. An emerging challenge is to further characterize activity levels and interactions of pathways to develop a deeper understanding of how specific alterations in these interactions promote tumor growth. *Drosophila* with its sophisticated genetics has proved valuable in studying cooperative oncogenesis. Using *Drosophila* tumor models, we report a tumor cell-specific network comprising four pathways. We show that Wingless and effector caspase Dronc direct a signal amplification loop involving JNK and Hippo-effector Yorkie (Yki). Our studies provide evidence from in vivo studies regarding the organization of tumor-promoting oncogenic pathways, which may be useful in developing precise and effective approaches for pathway inhibition. These pathways are evolutionarily conserved from flies to humans, suggesting that findings from flies can be extrapolated to mammalian cancers.

**Abstract:**

Cancer cells expand rapidly in response to altered intercellular and signaling interactions to achieve the hallmarks of cancer. Impaired cell polarity combined with activated oncogenes is known to promote several hallmarks of cancer, e.g., activating invasion by increased activity of Jun N-terminal kinase (JNK) and sustained proliferative signaling by increased activity of Hippo effector Yorkie (Yki). Thus, JNK, Yki, and their downstream transcription factors have emerged as synergistic drivers of tumor growth through pro-tumor signaling and intercellular interactions like cell competition. However, little is known about the signals that converge onto JNK and Yki in tumor cells and enable tumor cells to achieve the hallmarks of cancer. Here, using mosaic models of cooperative oncogenesis (*Ras^V12^*,*scrib^−^*) in *Drosophila*, we show that *Ras^V12^*,*scrib^−^* tumor cells grow through the activation of a previously unidentified network comprising Wingless (Wg), Dronc, JNK, and Yki. We show that *Ras^V12^*,*scrib^−^* cells show increased Wg, Dronc, JNK, and Yki signaling, and all these signals are required for the growth of *Ras^V12^*,*scrib^−^* tumors. We report that Wg and Dronc converge onto a JNK–Yki self-reinforcing positive feedback signal-amplification loop that promotes tumor growth. We found that the Wg–Dronc–Yki–JNK molecular network is specifically activated in polarity-impaired tumor cells and not in normal cells, in which apical-basal polarity remains intact. Our findings suggest that the identification of molecular networks may provide significant insights into the key biologically meaningful changes in signaling pathways and paradoxical signals that promote tumorigenesis.

## 1. Introduction

Over the past decade, genomic, transcriptomic, and proteomic data from several cancers have revealed the genetic alterations and changes in gene expression and epigenetic modifications linked to several cancers [1,2,3,4]. These approaches have revealed the extensive differences between normal and cancer cells, as well as the effects of cancer on multiple genes and signaling pathways [3]. Although these studies have provided valuable information about cancer cells, their origin, oncogenic processes, and signaling pathways, efforts now need to be directed at further characterizing tumors. For example, which proteins are expressed in the same cancer cell and are likely to physically interact? Which signaling pathways play decisive roles in promoting cell proliferation, survival, and metastasis in the different stages of cancer? Does the intricate distribution or clustering of ligand receptors affect cell–cell interactions [5]? For this, complementary approaches with in vivo models that allow manipulation of multiple genes (that represent the key cancer driver mutations) are required.

*Drosophila*, the common fruit fly, represents an efficient model organism to modulate the expression of multiple genes to study the signaling crosstalk that promotes tumor growth and progression [6,7]. Models of cooperative oncogenesis in *Drosophila*, such as clonal tumors induced by the activation of oncogenic Ras (*Ras^V12^*) in polarity-deficient *scribble (scrib)* [*Ras^V12^ scrib^−^*], *lethal giant larva (lgl)*, or *discs large (dlg)* mutant cells, show classic hallmarks of aggressive cancer growth exemplified by increased proliferation rate, reduced apoptosis and differentiation, and metastasis [8,9,10,11]. These models have been instrumental in establishing the links between the deregulation of cell polarity and increased signaling from Jun N-terminal kinase (JNK) and Yorkie (Yki), effectors of two key signaling pathways that regulate cell proliferation and apoptosis [12,13,14,15,16]. However, the upstream mechanisms by which JNK and Yki activation promote aggressive metastatic growth during oncogenic cooperation remain poorly defined.

We investigated the changes in signaling interactions to decipher how oncogenic cooperation [*Ras^V12^*,*scrib^−^*] promotes tumor growth. Herein, we show that the *Ras^V12^*,*scrib^−^* tumors grow aggressively by upregulating a previously unidentified molecular network comprising the Wingless (Wg, the *Drosophila* homolog of mammalian Wnt1), the effector caspase Dronc (*Drosophila* homolog of mammalian Caspase 9), JNK, and Yki (*Drosophila* homolog of Hippo effectors YAP/TAZ in mammals). The upregulation of these signals promotes the growth of *Ras^V12^*,*scrib^−^* tumors, and the depletion of these signals reverses these phenotypes. These signals are well-known for their roles in patterning and growth control [17,18,19,20,21] and intercellular interactions like cell competition [22,23,24,25,26,27,28]. We show that during oncogenic cooperation, these signals act together in a tumor cell-specific signaling network, in which Wg acts upstream of Dronc and regulates JNK and Yki. JNK activity changes from pro-apoptosis to pro-proliferation due to a paradoxical signaling switch that ultimately promotes Yki activity. We demonstrate that in *Ras^V12^*,*scrib^−^* tumor cells, Yki, in turn, upregulates JNK activity, causing the robust activation of both JNK and Yki. This bidirectional regulatory interaction results in the formation of a self-reinforcing JNK-Yki positive feedback signal-amplification loop downstream of Wg and Dronc. We show that this molecular network is sufficient to induce tumorigenesis in other contexts of oncogenic cooperation (such as *en* > *Yki*, *scrib^−^*, or *Ras^V12^//scrib^−^* [interclonal tumor model [10]]). Taken together, our data strongly support the functional significance of molecular networks rather than individual signals in cancer cells and provide novel insights into the molecular pathways and paradoxical signals that play a key role in tumor growth and progression.

## 2. Materials and Methods

### 2.1. Fly Stocks

The studies described here are conducted on *Drosophila melanogaster*, the common fruit fly. All flies were reared in a standard cornmeal–agar–molasses medium containing Tegosept and propionic acid. All fly stocks used in this study are described in Flybase (http://flybase.bio.indiana.edu) accessed on 1 March 2024.

The following stocks were used in this study: Canton-S, *FRT82B* (BL#5619), *FRT82B scrib^j7b3^* (DGRC#111422), *FRT82B scrib^2^* [29], *FRT82B scrib^2^ TubGAL-80* (this study), *diap1-lacZ* (BL#12093), *dronc^1.7kb^-lacZ* [30], *ex^679^-lacZ* [31], *wg-lacZ* [32], *AyGAL4, UAS-GFP* [33], *en-GAL4* (BL#1973), *UAS-GFP* (BL#1521), *FRT82B Ubi-GFP* (BL#5188), *FRT82B Tub-GAL80* (BL#5135), *MS1096-Gal4* (BL#8860), *UAS-P35* (BL#5072), *UAS-Ras^V12^* (BL#4847), *UAS-sgg^S9A^* (BL#5255), *UAS-Bsk^DN^* (BL#6409), *UAS-dronc^RNAi^* (gift from A. Bergmann, 8091R-1 from NIG-Fly (http://www.shigen.nig.ac.jp/fly/nigfly/index.jsp) accessed on 1 March 2024), *UAS-Yki^N+CRNAi^* [34], *UAS-proDronc* (gift from A. Bergmann), *UAS-jun^aspv^* [35], *UAS-Arm^S10^* (BL#4782), and *UAS-Yki* [36].

### 2.2. Generation of Somatic Clones

To make MARCM clones, we *crossed eyFLP or UbxFLP; AyGAL4 UASGFP; FRT82B Tub-GAL80* virgins with males of the appropriate genotypes. Heat shock-mediated “Flp-out” clones were generated by giving a 7 min heat shock at 37 °C to second instar larvae generated by crossing *AyGAL4 UASGFP* flies with *UAS Yki*. “Flp-out” clones were also generated by crossing *yw hsFLP*; *enGAL4 ex^697^-lacZ*; *FRT82B UbiGFP* flies with *UAS Yki*; *FRT82B scrib^2^* flies (for *en > Yki*; *scrib^−^* clones Figure 5). Interclonal oncogenic cooperation studies were performed by crossing *yw eyFLP*; *AyGAL4 UASGFP*; *FRT82B Tub-GAL80 scrib^2^* virgins with *yw hsFLP*; *+*; *UASRas^V12^ FRT82B* males. All experiments were performed at 25 °C unless otherwise specified.

### 2.3. Immunohistochemistry and Image Acquisition

Immunohistochemistry was performed following our published protocol [37]. Briefly, imaginal discs from wandering 3rd instar larvae were dissected in 1XPBS (phosphate buffered saline), fixed in 4% paraformaldehyde (PFA) for 20 min (min) at room temperature (RT), washed 3 × 10 min each in 1XPBST (1XPBS + 0.2% Triton X-100), and blocked in PBSTN (PBST+ 2% Normal donkey serum) for 1 h before incubation with primary antibody at 4 °C overnight. The samples were then washed 3 × 10 min each in 1XPBST, incubated in an appropriate secondary antibody for two hours at RT, washed in 1XPBST for 20 min, and mounted in Vectashield (Vector Laboratories Inc., Burlingame, CA, USA).

The following primary antibodies were used: rabbit anti-cleaved Caspase 3 (1:250), rabbit anti-DCP1 (1:100), and rabbit anti-pJNK (1:250) from Cell-Signaling Technology, Danvers, MA, USA, mouse anti-Wingless (1:100), mouse or rabbit anti-beta-galactosidase (1:100), mouse anti-MMP1 (1:250), and rat anti-E-Cadherin (dCAD2, 1:100) from DSHB, Iowa, IA, USA, guinea pig anti-Dronc (1:400, from Dr. H. D. Ryoo), mouse anti-DIAP1 (1:250, from Dr. B. Hayes), and rabbit anti-Yorkie (1:400, from Dr. K. Irvine). The secondary antibodies were donkey Fab fragments conjugated to Cy3 or Cy5 against rabbit, mouse, rat, or guinea pig hosts (Jackson ImmunoResearch Labs, West Grove, PA, USA).

Image acquisition: We used an Olympus Fluoview 1000 confocal microscope to acquire projections of *Z*-stacks of confocal images. To enable comparison between samples, all images in an experimental group were acquired at identical magnification and similar imaging conditions. The final figures were made using Adobe Photoshop CS6.

### 2.4. Statistical Analyses

Statistical analyses were performed using Microsoft Excel 2013. The magnetic lasso tool was used for clone size comparison (Figures 1A and 2A Wild type n = 11, rest n = 25). Mean pixel values of clone area were obtained using the Histogram function in Photoshop CS6 and analyzed using the non-parametric Mann–Whitney test (two-tailed), assuming statistical significance at *p* < 0.05. Signal intensity levels for both inside (GFP-positive) and outside (GFP-negative), the clones were obtained using the histogram function in Adobe Photoshop CS6 (n = 5). The average wild-type values were used to find the normalization factor to determine and calculate differences in intensity levels between wild-type, *scrib^−^*, *Ras^V12^*, and *Ras^V12^*; *scrib^−^* clones for all antibodies tested. Bar scatterplots were then generated to show fold-change in Dronc, Wg, *dronc^1.7kb^-lacZ*, *wg-lacZ*, *diap1-lacZ*, pJNK levels (Figure 1 and Appendix A), and DIAP1 (Figure 2F and Appendix A). Similarly, the normalization factor was calculated using wild-type values (GFP-negative areas) to assess changes in the expression of Wg (Figure 3), Yki, pJNK, and Dronc (Figure 4). Statistical significance (*p* < 0.05) was tested using a non-parametric Mann–Whitney test (two-tailed), and all graphs were plotted using GraphPad Prism9.0.

### 2.5. qRT-PCR

RNA was extracted from eye imaginal discs from wandering third instar larvae. At least 40–50 discs were collected per genotype in TRIzol. The ZYMO RNA Clean and Concentrator Kit was used for RNA extraction. To prepare cDNA, 200 ng/μL RNA/sample was reverse transcribed using the cDNA synthesis kit (Cytiva, Danaher Corporation, Washington, DC, USA). qRT-PCRs were performed in triplicate per sample using the iQ SYBER Green Supermix (Bio-Rad Laboratories, Hercules, CA USA) on an iCycler iQ™ Real-Time PCR Detection System (Bio-Rad Laboratories, Hercules, CA USA). Three biological and three technical repeats were performed for each experiment, and data were analyzed using the ddCt method.

The following primers (IDT) were used:

*wg*: Forward primer*: 5′* CGT CAG GGA CGC AAG CAT A-3′

Reverse Primer*: 5′* ATT GTG CGG GTT CAG TTG G-3′

*dronc*: Forward primer*: 5′* CGA TGG ATC TGT GGT CGA TAT G-3′

Reverse Primer*: 5′* GGC TTC GCT CGT CTT CTT TA-3′

*Gapdh*: Forward primer*: 5′* TAA ATT CGA CTC GAC TCA CGG T-3′

Reverse Primer*: 5′* CTC CAC CAC ATA CTC GGC TC-3′

Statistical analyses were performed using the non-parametric Mann–Whitney test (two-tailed), assuming statistical significance at *p ≤* 0.05. Graphs were plotted using GraphPad Prism9.0.

## 3. Results

### 3.1. Ras^V12^,scrib^−^ Cells Grow Robustly and Induce JNK, Yki, Dronc, and Wg

To investigate the changes in intercellular interactions that promote tumor growth, we made *ey-FLP*- and *Ubx-FLP*-induced MARCM clones [GFP positive] in the eye (Figure 1A–C) and wing discs (Figure 1D,E) respectively and monitored clone size to assess tumor growth (Figure 1A). Compared to wild-type (Figure 1A), *scrib^−^* clones grew poorly (Appendix A) [38]. Given that *scrib^−^* clones are competed out due to cell competition-mediated apoptosis [12,38], we blocked apoptosis by overexpressing *UASp35* (Appendix A), a pan-Caspase inhibitor [39]. The inhibition of apoptosis improved the growth of *scrib-* clones (*scrib^−^*, *p35*, Figure 1A and Appendix A); however, the discs remain monolayered, and clones did not form invasive tumors (Appendix A and YZ sections in Appendix A). In contrast, the coexpression of oncogenic Ras (*UASRas^V12^*) in *scrib^−^* cells (*Ras^V12^*,*scrib^−^*) resulted in robust aggressive and invasive tumors [8] that grew severalfold compared to wild-type, *scrib^−^*, or *scrib^−^*, *p35* clones (Figure 1A). These data suggest that a paradoxical switch changes the ability of *scrib^−^* cells, and inter-cellular interactions may play a critical role in these observed effects. Next, to understand the nature of the intercellular interactions that may underlie the differences in clone sizes, we tested cell death by using an antibody against activated Caspase 3 [Casp3*]. In comparison to wild-type clones (Figure 1B,B’), *Ras^V12^*,*scrib^−^* clones show cell death both inside and outside the clone, and interestingly, the bulk of the observed cell death was induced in wild-type cells surrounding the clone (Figure 1C,C’ and Appendix A). Similarly in the wing discs, compared to wild-type (Figure 1D), *Ras^V12^*,*scrib^−^* clones grew into large invasive tumors (Figure 1E and Appendix A). Taken together, these data suggested that the *Ras^V12^*,*scrib^−^* clones grow robustly. Therefore, as a next step, we tested if JNK, Yki, Dronc, and Wg, markers that have previously been linked to cell survival and proliferation during cell competition [22,23,24,40] were also affected in the *Ras^V12^ scrib^−^* tumors.

JNK and Yki are known to interact in a context-dependent manner [12,13,14,41], and increased JNK or Yki activity has been linked to tumor growth [12,42,43,44,45]. In contrast, the elimination of *scrib^−^* cells by cell competition involved JNK-dependent suppression of Yki activity [12,38,46,47,48]. To test Yki activity, we employed the Yki-reporter, *diap1-lacZ*, which is expressed ubiquitously in wild-type (Appendix A) discs. Increased Yki activity due to Yki overexpression in “Flp-out” (*AyGAL4 > Yki*) clones showed the robust induction of *diap1-lacZ* (Appendix A). In the *Ras^V12^*,*scrib^−^* clones, *diap1-lacZ* was strongly induced (Appendix A). Interestingly, *diap1-lacZ* was also induced in peri-tumoral cells (non-cell autonomously in wild-type cells) abutting the *Ras^V12^*,*scrib^−^* tumors (Appendix A). The normalized pixel intensity from the controls and experimental groups was quantified and showed a significant increase in *Ras^V12^*,*scrib^−^* clones (Appendix A). Compared to wild-type (Appendix A), pJNK was robustly induced in the *Ras^V12^*,*scrib^−^* clones (Appendix A, quantified in Appendix A). Thus, consistent with previous data, JNK and Yki activities were simultaneously upregulated in the *Ras^V12^*,*scrib^−^* tumors. We then investigated the signals that converge onto Yki and JNK to promote the growth of *Ras^V12^*,*scrib^−^* tumors.

Previous studies have shown that Dronc (the initiator caspase that induces apoptosis through activation of caspase 3) and Wg (the secreted ligand of the Wingless/Wnt pathway that acts as a mitogen) play critical roles in intercellular interactions like cell competition and compensatory proliferation [27,49,50,51]. Yki also transcriptionally regulates *dronc* and *wg* [52,53]. We tested levels of active Dronc using an antibody against Dronc^CA^, the activated form of Dronc [51], and Wg in the *Ras^V12^*,*scrib^−^* clones (Figure 1). Dronc is expressed ubiquitously in wild-type imaginal discs (Figure 1F,F’, quantified in Figure 1H), and its levels remained unaffected in *scrib^−^* cells in the wild-type background (Figure 1H). Therefore, wild-type Dronc levels may be sufficient for the elimination of *scrib^−^* cells in the presence of elevated pJNK levels. In comparison, Dronc was strongly upregulated in *Ras^V12^*,*scrib^−^* cells (Figure 1G,G’). Quantification showed that Dronc levels were significantly upregulated (1.7-fold) in *Ras^V12^*,*scrib^−^* cells compared to either wild-type or *scrib^−^* cells (Figure 1H). Taken together, these data suggested that Dronc plays a non-apoptotic role and promotes *Ras^V12^*,*scrib^−^* tumor growth. In wild-type eye discs, Wg is expressed at the lateral margins anterior to the morphogenetic furrow (Figure 1I,I’, quantified in Figure 1K) [32]. This pattern of Wg expression was unaltered in *scrib^−^* cells generated in the wild-type background (Figure 1K). In *Ras^V12^*,*scrib^−^* clones, Wg was robustly induced (Figure 1J,J’). When compared to wild-type or *scrib^−^* cells, Wg expression was significantly upregulated (2.7-fold) in *Ras^V12^*,*scrib^−^* cells (Figure 1K), suggesting that the steep upregulation of Wg levels in *Ras^V12^*,*scrib*^−^ may itself promote super-competition by promoting proliferation through its mitogenic functions.

Next, we checked if *dronc* and *wg* transcription was affected in *Ras^V12^*,*scrib^−^* cells using the *dronc^1.7kb^-lacZ* (Figure 1L–M’) and *wg-lacZ* (Figure 1O–P’) reporters, quantified in Figure 1N,Q, respectively. We found that compared to the wild-type (Figure 1L), *dronc^1.7kb^-lacZ* expression was robustly induced in the *Ras^V12^*,*scrib^−^* clones (Figure 1M,M’). Similarly, the normal pattern of *wg-lacZ* expression in eye discs (Figure 1O) was disrupted due to ectopic induction of *wg-lacZ* in *Ras^V12^*,*scrib^−^* clones (Figure 1O’). We confirmed the increased expression of *dronc* and *wg* in *Ras^V12^*,*scrib^−^* clones using qRT-PCR (Appendix A). The upregulation of *dronc* and *wg* in the tumor cells suggests that these genes promote tumor growth, possibly by co-opting compensatory mechanisms in which *dronc* plays a non-apoptotic role and *wg* plays a mitogenic role. Taken together, these data suggest that four signals associated with intercellular interactions (Yki, JNK, Wg, caspases) were upregulated in aggressively growing *Ras^V12^*,*scrib^−^* tumors.

### 3.2. Ras^V12^,scrib^−^ Tumors Require JNK, Yki, Dronc, and Wg for Growth

To understand if some or all of these signals played a key role in the aggressive growth of *Ras^V12^*,*scrib^−^* tumors, we tested their requirement by independently downregulating these signals in *Ras^V12^*,*scrib*^−^ clones (Figure 2). To achieve this, we used well-established transgenes, for example, RNAi, to knockdown *Dronc*, which impairs caspase-mediated signaling [*UAS-Dronc^RNAi^*] [52], the dominant-negative form of Bsk, which is a potent suppressor of JNK signaling [*UAS-Bsk^DN^*] [38], *UAS-Sgg^S9A^*, which dominantly blocks Wg signaling [54], and *UAS-Yki^N+CRNAi^*, which causes the inactivation of Yki-mediated signaling [34].

A comparison of clone sizes revealed that individual downregulation of the four signaling pathways significantly decreased the growth of the *Ras^V12^*,*scrib^−^* cells (Figure 2A–E). These findings suggested that the four signals are independently required for the aggressive growth of *Ras^V12^*,*scrib^−^* tumors. Therefore, we next tested if loss of cellular fitness or interactions amongst these signals was an underlying cause for the change in tumor size.

### 3.3. Downregulation of JNK, Yki, Dronc, and Wg Impairs Cellular Fitness

We hypothesized that the reduction in clone size was due to altered fitness within *Ras^V12^*,*scrib^−^* cells. First, we tested changes in DIAP1 levels (Figure 2B–E red, grey) in conditions where at least one of these four signals was downregulated (Figure 2B–F). As DIAP1 inhibits caspase activation, it protects cells from caspase-mediated apoptosis [55]. DIAP1 was ubiquitously expressed in wild-type cells (Appendix A), downregulated in *scrib^−^* cells (Appendix A, yellow arrowheads), and robustly induced in *Ras^V12^*,*scrib^−^* clones (Appendix A). However, when Yki (Figure 2B), JNK (Figure 2C), Dronc (Figure 2D), or Wg (Figure 2E) were downregulated in *Ras^V12^*,*scrib^−^* clones (Figure 2C–E, yellow outline in grey channels), DIAP1 was downregulated (quantified in Figure 2F). This finding suggests that *Ras^V12^*,*scrib*^−^ tumor cells had decreased survival and were susceptible to elimination by apoptosis (Figure 2B–E, blue, grey), when the tumor-promoting signals were downregulated. The downregulation of Yki (Figure 2B) or Wg (Figure 2E) resulted in apoptosis both inside (Figure 2B,E yellow arrowheads) and outside (Figure 2B,E, red arrowhead) the *Ras^V12^*,*scrib^−^* clones, suggesting that Wg and Yki protect *Ras^V12^*,*scrib^−^* tumors during competitive interactions. The downregulation of the JNK pathway (Figure 2C) or Dronc (Figure 2D) resulted in apoptosis at the clone boundary (Figure 2C,D, red arrowhead), but not within the clones, likely because these cell death regulators (JNK and Dronc) were suppressed in the clones. Taken together, these data suggested that the downregulation of the tumor-promoting signals impaired cellular fitness in *Ras^V12^*,*scrib^−^* cells.

These observations were further supported in control experiments, where we made MARCM clones in which we depleted each component of the molecular network in otherwise wild-type cells (Appendix A). Wg is a target of its own signaling [56,57,58]. Therefore, we made *UAS-Sgg^S9A^* control clones (Appendix A) and confirmed that the endogenous expression of Wg was suppressed in *UAS-Sgg^S9A^* clones using anti-Wg antibody (shown by the yellow arrow in Appendix A). The downregulation of Wg signaling (Appendix A red, Appendix A grey) caused cell death (Appendix A blue, Appendix A grey) and the downregulation of DIAP1 (Appendix A red, Appendix A grey). However, the levels of pJNK (Appendix A blue, Appendix A grey) remained unaltered. The depletion of Dronc blocked cell death (Appendix A blue, Appendix A grey) and did not significantly affect DIAP1 (Appendix A red, Appendix A grey), Wg (Appendix A red, Appendix A grey), or pJNK (Appendix A blue, Appendix A grey) expression. Blocking JNK signaling (Appendix A) showed similar effects to those of the downregulation of Dronc on DIAP1 (Appendix A red, Appendix A grey), cell death (Appendix A blue, Appendix A grey), Wg (Appendix A red, Appendix A grey), or pJNK (Appendix A blue, Appendix A grey). The depletion of Yki (Appendix A) resulted in the downregulation of DIAP1 (Appendix A red, Appendix A grey) and a strong non-cell autonomous induction of cell death (Appendix A blue, Appendix A grey). No significant effects on Wg (Appendix A red, Appendix A grey) or pJNK (Appendix A blue, Appendix A grey) were seen. Thus, Wg and Yki emerged as key signaling proteins among these four pathways.

### 3.4. Wg, Dronc, JNK, and Yki Form a Molecular Network in Ras^V12^,scrib^−^ Tumors

Next, we tested if Wg, Dronc, JNK, and Yki regulated one another to promote the aggressive growth of *Ras^V12^*,*scrib^−^* tumors (Figure 3). The downregulation of Wg signaling in *Ras^V12^*,*scrib^−^* cells (Figure 3A) resulted in small tumors and a reduction in Wg expression (Figure 3A red, A” grey). Interestingly, depleting Dronc (Figure 3B), JNK (Figure 3C), or Yki (Figure 3D) caused a significant decrease in clone size, but Wg was strongly induced in these clones (Figure 3B”–D”). It is notable that Wg is induced in both a cell-autonomous and non-cell-autonomous manner. Normalized signal intensity is quantified in Figure 3E. Thus, we identified that Wg acts upstream of Dronc, JNK, and Yki in the network of signals that promote *Ras^V12^*,*scrib^−^* tumor growth.

To test if these signals interact with each other, we checked the effects of Yki depletion (*Yki^RNAi^*; *Ras^V12^*,*scrib^−^*, Figure 4A–D) on Dronc (Figure 4A) and pJNK expression (Figure 4B). The depletion of Yki (confirmed in Figure 4C red, C’ grey) did not affect the induction of Dronc (Figure 4A blue, A’ grey); however, pJNK expression was downregulated in these clones (Figure 4B red, B’ grey). We quantified the normalized mean grey values and plotted the fold change comparing wild-type to *Ras^V12^*,*scrib^−^* and *Yki^RNAi^*; *Ras^V12^*,*scrib^−^*. Graphs show the significant downregulation of Yki and pJNK, whereas Dronc accumulation was not affected (Figure 4D). Next, we downregulated JNK signaling (Figure 4E–G) through the overexpression of a dominant negative form of basket (Bsk) in *Ras^V12^*,*scrib^−^* (*Bsk^DN^*; *Ras^V12^*,*scrib^−^*), which results in the downregulation of pJNK (Figure 4E blue, E” grey), but Dronc expression remained upregulated in these clones (Figure 4E red, E’ grey). Interestingly, Yki expression was also depleted in *Bsk^DN^*; *Ras^V12^*,*scrib^−^* cells (Figure 4F red, F’ grey). We plotted the normalized fold change in the expression of Dronc, Yki, and pJNK, which confirms these effects (Figure 4G). Interestingly, the downregulation of Dronc (*Dronc^RNAi^*; *Ras^V12^*,*scrib^−^*, Figure 4H–J) in *Ras^V12^*,*scrib^−^* cells (Figure 4H red, H’ grey) showed the downregulation of JNK (Figure 4H blue, H” grey) and Yki (Figure 4I red, I’ grey). Quantification shows that compared to wild-type or *Ras^V12^*,*scrib^−^*, the significant downregulation of Yki and JNK is seen in *Dronc^RNAi^*; *Ras^V12^*,*scrib^−^* clones (Figure 4J). The downregulation of Wg (Figure 4K–M) in *Ras^V12^*,*scrib^−^* cells (*Sgg^S9A^*; *Ras^V12^*,*scrib^−^*) showed the downregulation of Dronc (Figure 4K red, K’ grey), pJNK (Figure 4K blue, K” grey), and Yki (Figure 4L red, L’ grey). Quantification of the effects of downregulation of Wg revealed significant downregulation of Dronc, Yki, and pJNK compared to *Ras^V12^*,*scrib^−^* clones (Figure 4M). These data show that Dronc levels remain upregulated and comparable to *Ras^V12^*,*scrib^−^* cells in combinations in which either Yki (Figure 4D) or JNK (Figure 4G) are downregulated. Thus, these data suggest that Dronc acts upstream of Yki and JNK in this signaling network. Furthermore, JNK and Yki regulate each other, suggesting that they may act in a feedforward loop downstream of Dronc. Overall, our data suggest that the growth of the *Ras^V12^*,*scrib^−^* clones depended on a network involving Wg-dependent activation of Dronc, which controls a JNK-Yki mediated signal amplification loop that sustains high levels of JNK and Yki activities.

### 3.5. Cooperative Interactions Stimulate the Molecular Network and Tumor Growth

The preceding data led us to ask if oncogene activation, loss of polarity (*scrib^−^*), or cooperative interactions were important in stimulating the molecular network and tumor growth. The loss of polarity (*scrib*) alone is insufficient to induce aggressive growth in somatic clones (Figure 1 and Appendix A) [8,9,12,38]; therefore, we tested the importance of cooperative interactions on the induction of the Wg–Dronc–JNK–Yki network and tumor growth. We tested two scenarios in which the loss of polarity could synergize with oncogene activation.

In the first scenario, we generated *scrib^−^* clones [GFP-negative clones in the *FRT82B Ubi-GFP* background generated by *hs-FLP*] in wing discs in which Yki was overexpressed [*UAS-Yki*] in the posterior compartment using *en-GAL4* [*en > Yki*; *scrib^−^*] (Figure 5). In the eye or wing discs, *scrib* mutant clones are susceptible to elimination by cell competition due to the upregulation of JNK (Appendix A), downregulation of Yki activity, as seen through the suppression of *diap1-lacZ* (Appendix A yellow arrowheads), and little to no effect on Dronc (Appendix A) or Wg (Appendix A) expression. Whereas, consistent with published data, the over-expression of Yki results in robust overgrowth due to the induction of Yki activity, leading to the upregulation of JNK [14,16], activation of DIAP1 (Appendix A), ectopic induction of Wg [16], and suppression of Dronc [52]. Thus, the loss of *scrib* had clear and distinguishable effects from Yki overexpression (Figure 5 and Appendix A) on all four signals. These distinct effects allowed an ideal opportunity to test altered signaling caused during oncogenic cooperation by Yki overexpression in *scrib* mutant cells. We found that *scrib^−^* clones grew to significantly larger sizes in the posterior (P) compartment in which Yki was overexpressed (Figure 5A–E) and showed reduced E-cadherin expression (Figure 5A, red, grey, red asterisk). Increased Yki activity in the posterior compartment was monitored by *ex-lacZ* expression (Figure 5B red, B’ grey), which also allows marking the anterior–posterior (A/P) boundary (shown in the yellow dotted line in Figure 5). In addition to the increased *ex-lacZ* staining observed in the posterior compartment due to *yki* overexpression, *Yki* activity in the *scrib-* cells in the posterior compartment was very strongly induced within and around the clones (Figure 5B, outlined by the yellow line). We called this dramatic increase in Yki activity “super-induction”. In these clones, pJNK (Figure 5C red, C’ grey), Wg (Figure 5D red, D’ grey), and MMP1 (Figure 5E red, E’ grey) were induced in and around the *scrib^−^* clones, potentially leading to the establishment of the JNK–Yki loop and invasiveness by the induction of MMP1. The Yki, pJNK, and Wg signals spread to several cells outside the *scrib^−^* clones, with Yki activity propagating the farthest (Figure 5B,B’). In contrast, the anterior (A) compartment *scrib^−^* clones (Figure 5B–E blue outline) behaved like clones in the wild-type background (Appendix A) and were eliminated through cell competition. A comparison of the P and A compartment-specific clones suggested that the JNK–Yki signal amplification loop was activated specifically in the tumor cells in the P compartment in which cooperative interactions occurred due to Yki expression in *scrib^−^* cells.

In the second scenario, we tested if the Wg–Dronc–JNK–Yki network was activated during interclonal cooperation when *Ras^V12^* and *scrib^−^* cells were adjacent to each other (*Ras^V12^//scrib^−^*) [10]. Consistent with previous data, aggressive *Ras^V12^* tumors (GFP-positive) were formed adjacent to smaller *scrib^−^* clones (GFP negative, Figure 6A) [10]. Interestingly, the Ras*^V12^* clones extruded apically and showed a multi-layered invasive phenotype (Figure 6A–A”, Y-Z sections in Figure 6B–B”). The *Ras^V12^* clones showed robust growth despite the induction of cell death at the clone boundary between *Ras^V12^* (GFP-positive) and *scrib^−^* (GFP-negative) (Figure 6C,C’) clones. The expression of Wg was robustly induced in the *scrib^−^* clones (Figure 6D,D’), whereas pJNK levels appear upregulated in both *scrib^−^* and *Ras^V12^* clones (Figure 6E,E’) [10,38]. Yki expression showed higher nucleocytoplasmic distribution in the *Ras^V12^* clones (Figure 6F,F’). Taken together, these data suggest that the robust growth of the *Ras^V12^* clones was dependent on the interclonal signaling interactions with *scrib^−^* clones and involved caspases, Yki, Wg, and JNK signaling. Overall, these two experimental approaches reaffirmed that the induction of the molecular network was intimately linked to tumor growth in diverse scenarios.

### 3.6. The Role of Apical–Basal Polarity in the Establishment of Tumor-Specific Molecular Networks

The Wg–Dronc–Yki–JNK network requirement in promoting tumor growth under various cooperative contexts raised the question if these four signals were sufficient to induce tumors in normal cells or if the loss of polarity is essential to form aggressive tumors. To test this, we first tested the effect of overexpressing Wg–Dronc–Yki–JNK in normal cells using the Gal4-UAS system in wing-imaginal discs. We used the wing hinge and pouch-specific *MS1096-Gal4* (Figure 7A) to drive the expression of transgenes overexpressing Yki (*UAS-Yki*), *Dronc* (*UAS-proDronc*), *JNK* (*UAS-jun^aspv^*), and Wg (UAS-*Arm^S10^*) [MS1096 > *Yki*,*pro-Dronc*, *jun^aspv^*, *Arm^S10^*], which would result in the activation of all four signals (Figure 7). In wild-type wing discs, DIAP1 (Figure 7B) and pJNK (Figure 7E) are ubiquitously expressed, whereas Wg is expressed in the wing hinge, wing margin, and notum (Figure 7D). Wild-type discs show few randomly dying cells marked by DCP-1 (Figure 7C). Interestingly, the coexpression of these transgenes resulted in hyperplasia of the wing pouch and hinge region (Figure 7F,G). We observed moderate upregulation of DIAP1 in the wing pouch region (Figure 7F’) and increased cell death in the wing hinge region (Figure 7F”). Both Wg (Figure 7G,G’) and pJNK are induced in a patchy pattern in the dorsal hinge region (Figure 7G,G”). In control experiments, the overexpression of individual transgenes revealed that the overexpression of *Yki* (*MS1096 > Yki*) is capable of driving hyperplasia and the upregulation of DIAP1, Wg, and pJNK (Appendix A). The overexpression of pro-Dronc (*MS1096 > proDronc*) did not affect disc growth very strongly and showed moderate upregulation of DIAP1, cell death, Wg, and pJNK (Appendix A), suggesting that pro-Dronc plays a role in promoting survival, but not cell proliferation. Overexpression of *jun^aspv^* (*MS1096 > jun^aspv^*) showed the most interesting effects by reducing the size of the wing pouch due to strong suppression of DIAP1 and the induction of cell death (Appendix A). Although Wg and pJNK are induced in this combination (Appendix A), the excessive cell death in the wing pouch does not support the growth of these discs. The activation of the Wg pathway through the overexpression of Arm^S10^ (*MS1096 > Arm^S10^*) resulted in the upregulation of DIAP1, Wg, and pJNK (Appendix A). In addition, mild to moderate levels of cell death were observed in the wing pouch (Appendix A). Taken together, these data suggest that although the coactivation of these transgenes results in increased growth or mild hyperplasia, the presence of normal wild-type cells with intact polarity does not allow for the establishment of the network that would drive the tumor growth.

## 4. Discussion

Oncogenic cooperation is a key mechanism for tumor development and progression. Cooperative interactions between oncogenic *Ras* and loss of *scrib^−^* (*Ras^V12^*,*scrib^−^*) have been elegantly modeled using in vivo mosaic tumor models in *Drosophila* [8,9,12,38] and multiple mammalian cancer models [59,60,61,62,63]. We investigated how altered signaling and cell–cell interactions promote tumorigenesis during oncogenic cooperation. Previously ectopic upregulation of a network of transcription factors, e.g., Upd, JAK-STAT, AP-1, Myc, Ftz, Ets, Irbp18, Xrp1, slow border, and Vrille, were reported to promote the growth and invasiveness of the *Ras^V12^*,*scrib^−^* tumors [64,65,66,67,68]. Data acquired from transcriptomics or RNA-seq have shown the presence of multiple transcription factors in the same genotype suggesting that several pathways and signals may be simultaneously active that culminate in the upregulation of several different transcription factors in tumor cells. These studies also confirm that several transcription factors are active based on the activation of their target genes. However, the potential molecular networks established remained unclear. Here we present evidence to show that a tumor cell-specific network is formed in *Ras^V12^*,*scrib^−/−^* tumors in eye imaginal discs. This network is comprised of Yki, JNK, Wg, and caspases that act to promote fitness and aggressive growth (Figure 1 and Appendix A). Our findings from the eye discs are relevant to other epithelia, like the wing discs as the tumor-specific network, is recapitulated in the *en > Yki*; *scrib^−^* wing tumors (Figure 5).

Of these signals, JNK is very well documented as a modulator of Yki activity in the context of cell competition, compensatory proliferation, regeneration, and neoplastic tumors [12,14]. JNK is a pivotal stress-responsive kinase that promotes malignant transformation and metastasis of tumors [38,45]. JNK has also emerged as a key paracrine signal that links apoptosis to carcinogenesis [12]. In *scrib^−^* cells, JNK signaling suppresses Yki activity and promotes cell competition [12,69]. Concomitantly, JNK signaling causes the non-cell-autonomous propagation of Yki in the cells surrounding *scrib*^−^ clones and promotes compensatory proliferation [12,69]. Thus, in *scrib^−^* clones in the wild-type background, JNK (Appendix A) and Yki (Appendix A) activities are induced in two distinct cell populations, and wild-type levels of Dronc (Appendix A) or Wg (Appendix A) are sufficient to support the cell competition-mediated elimination of *scrib^−^* cells. Work from our lab and others showed that the suppression of cell death in *scrib^−^* cells led to increased proliferation and the loss of differentiation, but not tumorigenesis (Appendix A) [70,71]. In contrast, we found that these signaling interactions are significantly altered in *Ras^V12^*,*scrib^−^* cells, as Wg, Dronc, JNK, and Yki are all robustly upregulated (Figure 1 and Appendix A), suggesting a non-apoptotic tumor growth-promoting role for JNK and Dronc and growth-promoting mitogenic roles for Yki and Wg. Using reporter assays for *dronc* and *wg*, we found that *Ras^V12^*,*scrib^−^* cells show increased transcription of *dronc* (Figure 1L–N) and *wg* (Figure 1O–Q), which was confirmed using qRT-PCR (Appendix A). In addition, increased JNK activity using phospho-specific JNK antibody and Yki activity by *diap1-lacZ* reporter were confirmed in the *Ras^V12^*,*scrib*^−^ clones (Appendix A). These findings are consistent with reports that JNK signaling activity is converted from anti- to pro-tumor growth through the downregulation of Hippo signaling and ultimately leads to Yki activation [14,41]. Both Yki/YAP and JNK are linked to Wg signaling, as Wg is induced in a JNK-dependent manner during regenerative growth, tumorigenesis, and compensatory proliferation [51,72,73] and interacts with Yki and the Hippo pathway during organ development and tumorigenesis [74,75,76]. In addition, dysregulation of the Hippo, JNK, or Wg pathway is also linked to the activation of caspase-mediated apoptosis, and recently, mild caspase induction was shown to promote tumor growth [77,78,79]. Thus, the identification of this molecular network is significant in the context of inter-cellular interactions that promote tumor growth.

Previous studies have shown a role for the JNK and Hippo pathway in *Ras^V12^ scrib^−^* clones [38,41]. In our study, an assessment of the roles of JNK, Yki, Dronc, and Wg in *Ras^V12^*,*scrib^−^* tumorigenesis revealed several interesting insights. First, we found that JNK, Yki, Dronc, and Wg were all required for the aggressive growth of *Ras^V12^*,*scrib^−^*-induced tumors, as the downregulation of these signals individually resulted in a significant reduction in tumor growth (Figure 2A). Second, we observed that in the absence of tumor-promoting signals, *Ras^V12^*,*scrib^−^*-induced tumors show reduced survival and fitness (Figure 2). Third, we identified that the four tumor-promoting signals form a tumor cell-specific signaling module in which Wg acts upstream of Dronc, which, in turn, acts upstream of a JNK–Yki-mediated positive feedback signal amplification loop (Figure 3 and Figure 4). Signal amplification of Yki and JNK activities caused by the JNK–Yki positive feedback loop plays a key role in promoting tumorigenesis in *Ras^V12^*,*scrib^−^* cells. Fourth, we confirmed that in other instances of oncogenic cooperation (*Ras^V12^//scrib^−^*), this signaling module can be recapitulated (Figure 5 and Figure 6). Fifth, the upregulation of JNK and Yki in normal epithelial cells with intact polarity is not sufficient to induce this network and tumor growth (Figure 7 and Appendix A), which is consistent with other instances of developmental interactions between Yki and JNK in polarized epithelia [80,81]. Taken together, our studies reveal that increased cellular fitness promoted by a molecular network comprising Wg, Dronc, JNK, and Yki may indeed be a mechanism co-opted for aggressive tumor growth. Furthermore, the overexpression of pathway components in the normal epithelia of wing discs revealed that the signal overactivation levels and the apical-basal polarity context are extremely important. This is because mild hyperplasia can occur by driving the individual transgenes, but robust overgrowth is not seen in cells with intact polarity. Further, for the JNK pathway, a threshold is critical, as stress-induced cell death occurs both when levels of JNK reach above or below the homeostatic levels (Figure 7 and Appendix A).

Another possible mechanism is the differential success of cancer and neighboring normal cells in competing for survival and other extracellular signals in terms of actively internalizing or limiting the extracellular spread of critical signals. In this context, we observed steep differences in the non-cell-autonomous spread of Yki activity among *scrib^−^* (Figure 5 and Appendix A), *Ras^V12^ scrib^−^* (Appendix A), and *en > Yki*; *scrib^−^* (Figure 5B) cells. Non-cell-autonomous Yki activity spreads to 3–5 cells in *scrib^−^* cells, several cells (8–10) in *en > Yki*; *scrib^−^*, and only a few cells (~2) around the *Ras^V12^*,*scrib^−^* clones. Further, cell-autonomous and non-cell-autonomous cell death is observed on the depletion of Yki (Figure 2B), suggesting that cells with impaired Yki expression are vulnerable to elimination by unidentified mechanisms. Interestingly, both cell-autonomous and non-cell-autonomous expression of Wg is seen when the components of the Wg–Dronc–JNK–Yki network are perturbed (Figure 3). Thus, limiting the extracellular spread of key signaling components like Yki or Wg may impact the growth potential of cancer cells. The mechanisms by which cancer cells limit the spread of these signals should be elucidated in future studies and may provide molecular insights about how benign and malignant tumors behave.

The biological significance of our findings is validated by other studies that show the formation of context-dependent Yki/YAP-mediated signaling loops as a bona fide mechanism for tumor growth [82,83]. YAP is not only regulated by signaling pathways like Wnt, TGFβ, and notch, but YAP can also collaborate with key cancer pathways to form transcriptional complexes that alter transcriptional programs, specifically in cancer cells [84,85]. At least three different mechanisms have been reported. First, YAP and its cognate transcription factors like TEAD control transcription by binding to promoters of target genes. Second, YAP collaborates with other transcription regulators to alter gene expression, for example, in colon cancer cell lines YAP1, β-catenin, and TBX5 transcriptional complex regulate (TCF- or TEAD-independent) target genes [86]. Third, YAP/TEAD binds distal regulatory elements to regulate the transcription of target genes [87,88,89]. ChIP-seq and deep-sequencing studies in cancer and normal cells showed that several YAP-binding regions also show a consensus motif for AP-1 transcriptional factors, suggesting that YAP/TEAD and AP-1 cooperatively regulate target genes representing a cross-talk between YAP and JNK signaling [88,89].

These findings are especially interesting considering our identification of the tumor-cell-specific Wg–Dronc–JNK–Yki molecular network and the JNK–Yki positive feedback signal amplification loop in the *Ras^V12^ scrib^−^* cells. Like YAP, *Drosophila* Yki is known to regulate transcription by binding to the promoter regions of genes [90]. Similarly, Yki can form transcriptional complexes that cooperatively alter gene expression in cancer cells, for example, Yki and the Ecdysone receptor coactivator Taiman were shown to alter the transcriptional output of Yki-inducible Taiman-dependent genes in cells with hyperactive Yki and neoplastic tumor growth [91]. Based on our data, it is possible that under conditions of oncogenic cooperation, Yki and the *Drosophila* AP-1 transcription factors may (cooperatively) bind other regulatory regions to drive altered transcriptional output in cancer cells. This conclusion is further strengthened by the finding that dFos, a component of the *Drosophila* AP-1 (*Drosophila* Jun and Fos heterodimer) transcription factor, is required for the JNK-mediated invasiveness of *Ras^V12^*,*scrib^−^* tumors [44,66]. In summary, cancer cells may establish specific molecular networks that promote tumor growth and metastasis, and it may be critical to understand these interactions to understand how oncogenic pathways are activated.

## 5. Conclusions

Our data are significant in light of emerging data from cancer genome studies that revealed that somatic mutations accrued by precancerous cells essentially activate hallmark cancer pathways that converge on a small number of protein complexes and signaling cascades [6]. Our genetic analyses indicate the hierarchy of one such signaling interaction, with Wg acting upstream of Dronc, JNK, and Yki, and in the future, it will be interesting to identify the molecular mechanisms underlying the establishment and regulation of this module. These approaches can also unveil targets for the modulation of the tumor and provide insights into the biomechanics of tumor progression. Our approach of modeling altered signaling interactions in a genetically tractable model is a powerful way to reconstruct key biologically meaningful changes in signaling pathways in cancer cells and provides a functional framework to study the changes in signaling pathway interactions that will generate new insights on mechanisms that promote tumor growth and progression.

## Figures and Tables

**Figure 1 cancers-16-01768-f001:**
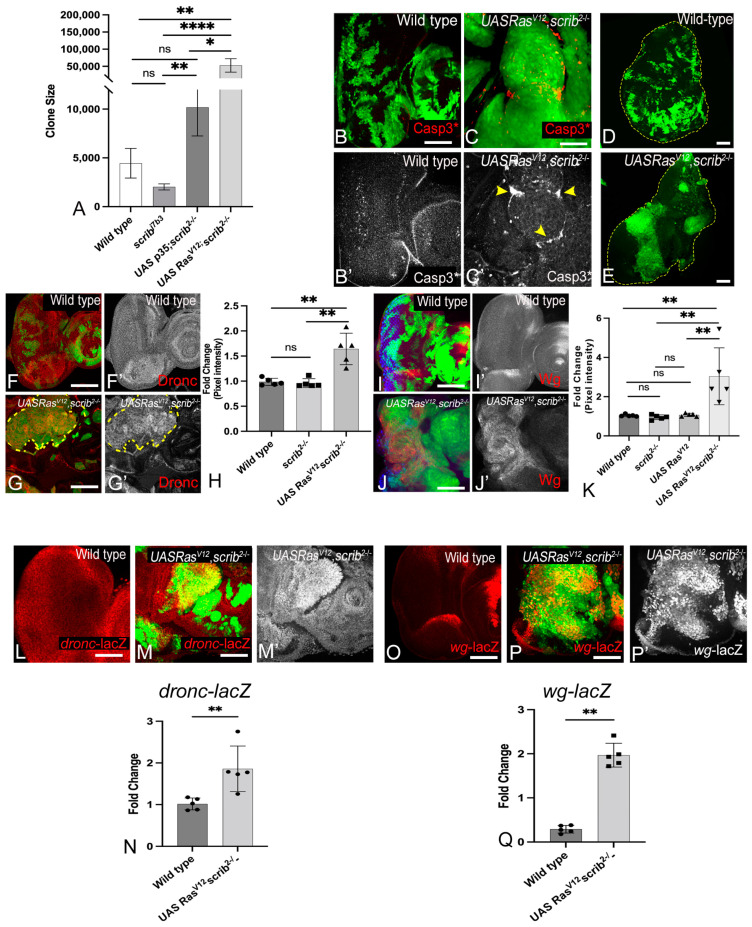
*Ras^V12^ scrib^−^* cells grow robustly and induce Dronc and Wg. (**A**) Graph shows quantification of clone size comparing the growth of wildtype clones to *scrib^−^*, *p35*; *scrib^−^*, and *Ras^V12^ scrib^−^* clones (n = 25). Panels show MARCM clones (GFP, green) from third-instar eye-antennal imaginal discs of (**B**) Wild-type, (**C**) *Ras^V12^ scrib^−^*, and wing discs from (**D**) wild-type and (**E**) *Ras^V12^ scrib^−^* respectively. Expression of cleaved caspase 3 [Casp3*] antibody is shown in the eye discs (red in **B**,**C**, grey in **B’**,**C’**). Yellow arrowheads mark dying cells in (**C’**). The outline of the wing disc is marked by a yellow dashed line in **D**,**E**. (**F**–**H**) Panels show Dronc expression (red, grey) in MARCM clones (green) induced in eye-antennal imaginal discs from (**F**,**F’**) wild-type, and (**G**,**G’**) *Ras^V12^ scrib^−^*. (**H**) Bar scatter plot shows quantification of Dronc expression in MARCM clones in the indicated genotypes (n = 5). (**I**–**K**) Panels show expression and quantification of Wingless expression in eye discs from (**I**,**I’**) wild-type and (**J**,**J’**) *Ras^V12^ scrib^−^* MARCM clones (green). (**K**) Bar scatter plot shows change in Wg expression in the indicated genotypes (n = 5). (**L**–**N**) *dronc^1.7kb^-lacZ* reporter expression (red, grey) in (**L**) wild-type and (**M**,**M’**) *Ras^V12^ scrib^−^* MARCM clones (green), quantified in (**N**). (**O**–**Q**) *wg-lacZ* reporter expression (red, grey) in (**O**) wild-type and (**P**,**P’**) *Ras^V12^ scrib^−^* MARCM clones (green), quantified in (**Q**). In all graphs, error bars show standard error of means (SEM). Statistical significance was calculated using Mann-Whitney test, where ns denotes *p*-value ≥ 0.05, and asterisks show significant interactions * *p*-value ≤ 0.05, ** *p*-value ≤ 0.01, and **** *p*-value ≤ 0.0001. Disc orientation is identical in all panels. Scale bar = 25 µM.

**Figure 2 cancers-16-01768-f002:**
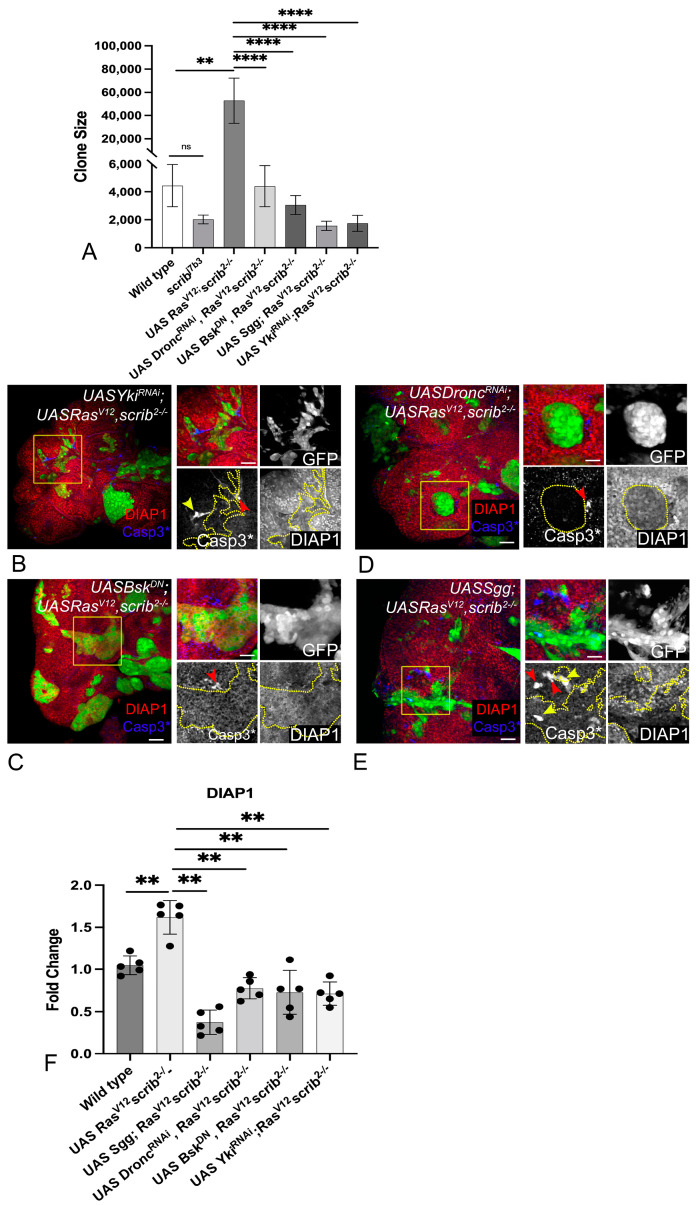
*Ras^V12^ scrib^−^* tumors require Dronc, Wg, JNK, and Yki for growth. (**A**) Bar graph shows quantification of clone size of the following genotypes: wild-type, *scrib^−^*, *Ras^V12^ scrib^−^*, *UASDronc^RNAi^ UASRas^V12^ scrib^−^*, UAS*Bsk^DN^ UASRas^V12^ scrib^−^*, *UASSgg^S9A^ UASRas^V12^ scrib^−^*, and *UASYki^N+CRNAi^ UASRas^V12^ scrib^−^* (n = 25) (**B**–**E**) Panels show DIAP1 (red) and Casp3* (blue) expression in eye imaginal discs containing MARCM clones (green) of the following genotype: (**B**) *UASYki^N+CRNAi^ UASRas^V12^ scrib^−^*, (**C**) *UASBsk^DN^ UASRas^V12^ scrib^−^*, (**D**) *UASDronc^RNAi^ UASRas^V12^ scrib^−^*, and (**E**) *UASSgg^S9A^ UASRas^V12^ scrib^−^*. Clone areas highlighted by yellow boxes are further magnified in the image to the right, and split channel images in greyscale are presented for each genotype. Clone outlines are marked by the yellow dotted line, red arrowheads highlight Casp3* positive apoptotic cells outside the clones, whereas yellow arrowheads mark Casp3* positive apoptotic cells inside the clones. (**F**) Bar scatter plot presents the fold change in DIAP1 levels comparing wild-type and *UASRas^V12^ scrib^−^* to all tested genotypes (n = 5). In all graphs, error bars show standard error of means (SEM). Statistical significance was calculated using Mann-Whitney test, where ns denotes *p*-value ≥ 0.05, and asterisks show significant interactions ** *p*-value ≤ 0.01, and **** *p*-value ≤ 0.0001. Disc orientation is identical in all panels. Scale bar = 25 µm.

**Figure 3 cancers-16-01768-f003:**
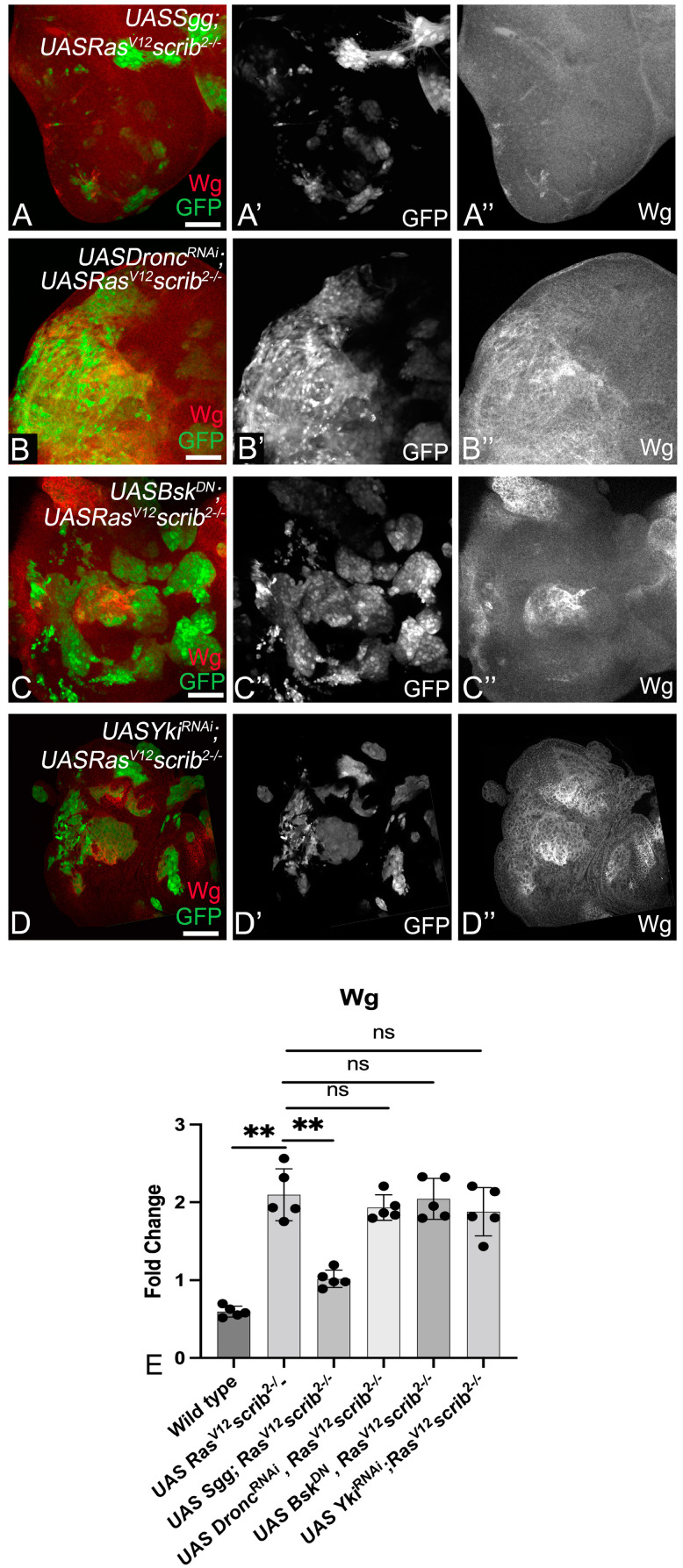
Wg acts upstream of the molecular network. Panels show Wg expression (red in **A**–**D**, grey in **A”**–**D”**) in MARCM clones (green in **A**–**D**, grey in **A’**–**D’**) of the following genotype: (**A**) *UASSgg^S9A^ UASRas^V12^ scrib^−^*, (**B**) *UASDronc^RNAi^ UASRas^V12^ scrib^−^*, (**C**) *UASBsk^DN^ UASRas^V12^ scrib^−^*, and (**D**) *UASYki^N+CRNAi^ UASRas^V12^ scrib^−^*. (**E**) Bar scatter plot presents the fold change in Wg levels comparing wild-type and *UASRas^V12^ scrib^−^* to all tested genotypes (n = 5). Error bars show standard error of means (SEM). Statistical significance was calculated using Mann-Whitney test, where ns denotes *p*-value ≥ 0.05, and asterisks show significant interactions ** *p*-value ≤ 0.01. The magnification and orientation of discs are identical in all panels. Scale bar = 25 µm.

**Figure 4 cancers-16-01768-f004:**
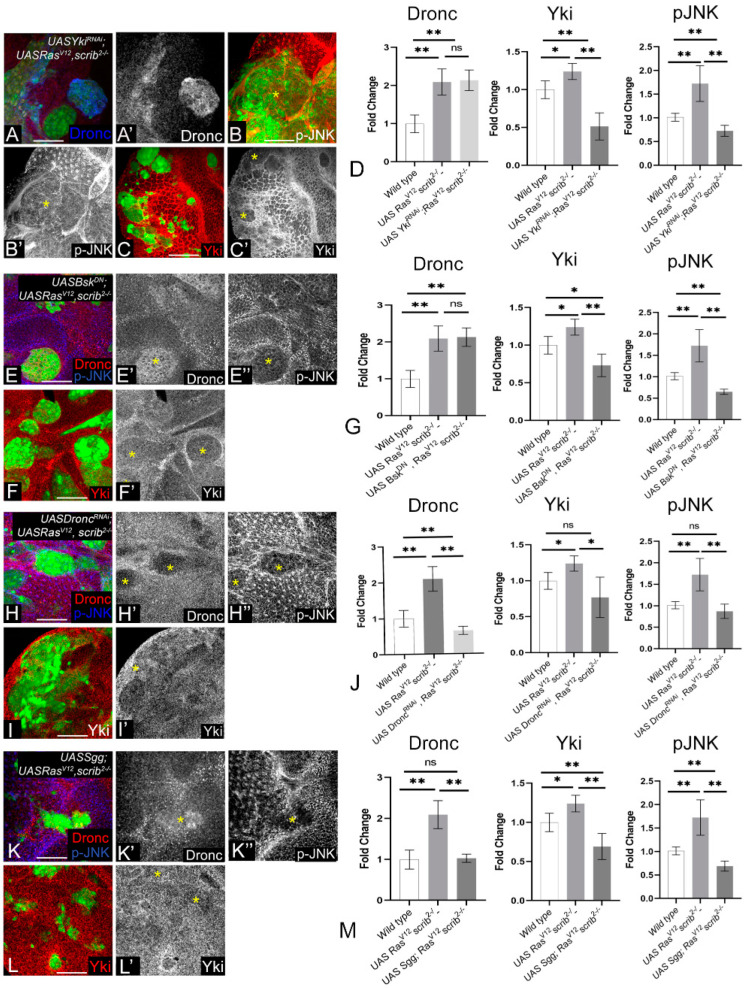
Wg, Dronc, JNK, and Yki form a molecular network in *Ras^V12^ scrib^−^* clones. Panels show Dronc, Yki, and p-JNK expression in MARCM clones (green) when the molecular network is perturbed. (**A**–**D**) Panels show *UASYki^N+CRNAi^ UASRas^V12^ scrib^−^* clones stained for Dronc ((**A**) blue, (**A’**) grey), pJNK ((**B**) red, (**B’**) grey) and Yki ((**C**) red, (**C’**) grey), quantified in (**D**). (**E**–**G**) *UASBsk^DN^ UASRas^V12^ scrib^−^* clones showing expression of Dronc ((**E**) red, (**E’**) grey), pJNK ((**E**) blue, (**E”**) grey), and Yki ((**F**) red, (**F’**) grey), quantified in (**G**). (**H**–**J**) Confocal images of *UASDronc^RNAi^ UASRas^V12^ scrib^−^* clones showing expression of Dronc ((**H**) red, (**H’**) grey), pJNK ((**H**) blue, (**H”**) grey), and Yki ((**I**) red, (**I’**) grey) are presented. The quantification of fold change is shown in (**J**). (**K**–**M**) Panels show confocal scans of *UASSgg^S9A^ UASRas^V12^ scrib^−^* clones stained for Dronc ((**K**) red, (**K’**) grey), pJNK ((**K**) blue, (**K”**) grey), and Yki ((**L**) red, (**L’**) grey), quantified in (**M**). The yellow asterisks mark representative MARCM clones in the grey channels. The quantifications in (**D**,**G**,**J**,**M**) show fold change comparison of normalized signal intensity between indicated genotypes, and error bars show SEM (standard error of means). Statistical significance was calculated using Mann-Whitney test, where ns denotes *p*-value ≥ 0.05, and asterisks show significant interactions * *p*-value ≤ 0.05, ** *p*-value ≤ 0.01. Disc orientation is identical in all panels. Scale bar = 25 µm.

**Figure 5 cancers-16-01768-f005:**
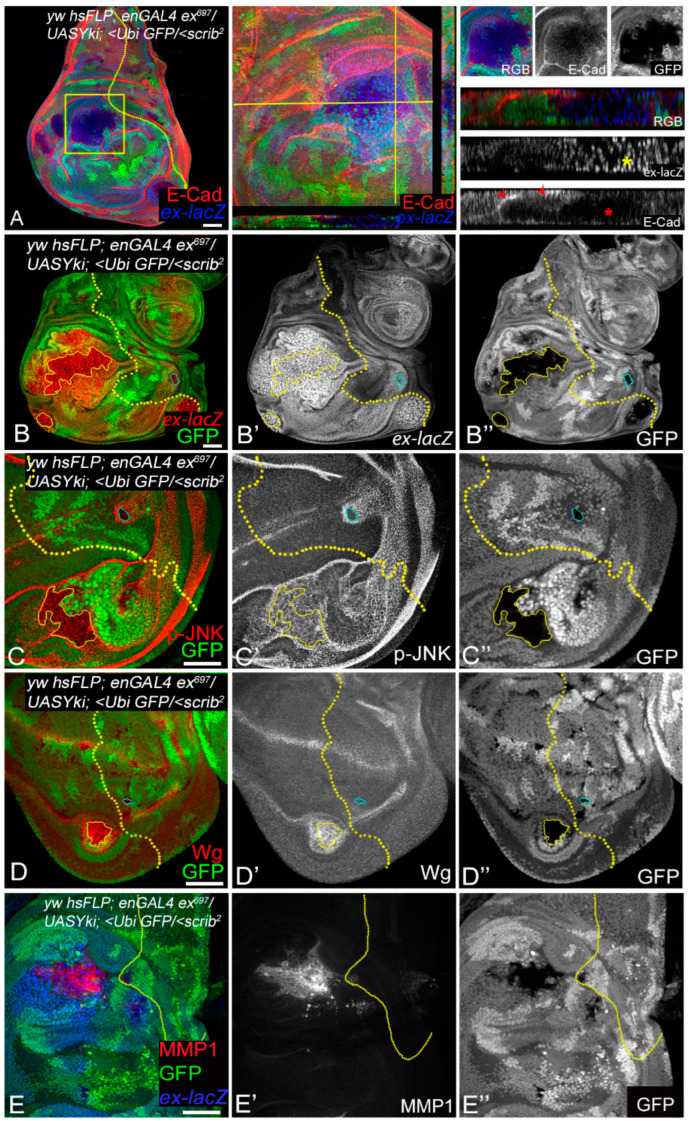
Cooperative interactions induce the Wg-Dronc-JNK-Yki network to stimulate tumor growth. (**A**) Wing imaginal discs showing *en > Yki*, *scrib^−^* clones (GFP-negative) stained for *ex-lacZ* (using antibodies to β-gal, (**A**) blue) and E-cadherin (anti-dCAD2 antibodies, (**A**) red). The area highlighted by the yellow box is shown at higher magnification in the panels to the right. A cropped image of the clone is shown next to depict changes in polarity and growth. Yellow asterisks show the multi-layered appearance of nuclei positive for *ex-lacZ* within the *en > Yki*, *scrib^−^* clones, and red asterisks point toward the loss of E-cadherin in the XZ section (along the plain marked in the middle panel), and normal E-cadherin localization is highlighted by red arrows for comparison. (**B**) Yki activity as assessed by β-gal expression ((**B**) red, (**B’**) grey) in *en > Yki*, *scrib^−^* clones is shown. (**C**–**E**) Wing discs containing *en > Yki*, *scrib^−^* clones (GFP negative) were assayed for ((**C**) red, (**C’**) grey), ((**D**) red, (**D’**) grey) or MMP1 ((**E**) red, (**E’**) grey) expression. The *scrib^−/−^* clones in posterior compartment (solid yellow line) and anterior compartment (blue lines) are highlighted for each marker. In all images, the anterior–posterior compartment boundary is marked by a yellow dotted line drawn based on *ex-lacZ* expression. Images in (**A**,**B**) are at 20× magnification, and (**C**–**E**) are at 40× magnification. Posterior is to the left and dorsal is up in all discs. Scale bar = 25 µm.

**Figure 6 cancers-16-01768-f006:**
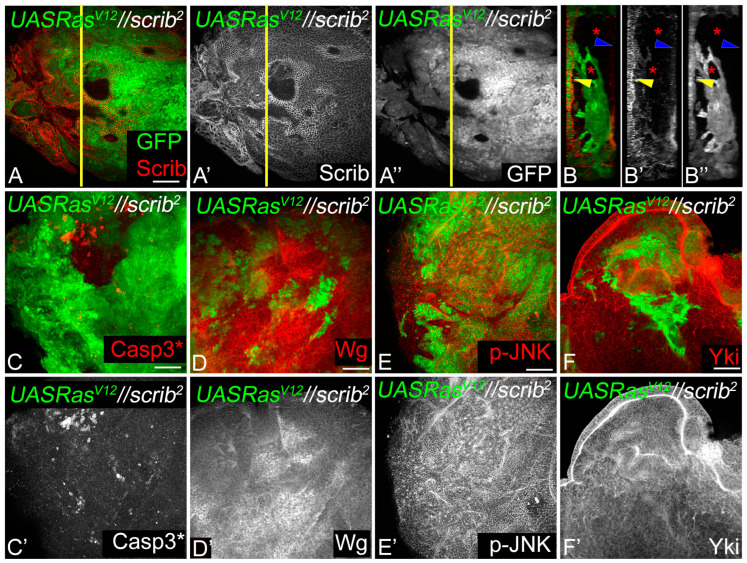
Wg-Dronc-JNK-Yki network drives tumor growth during interclonal interactions. (**A**) *Ras^V12^//scrib^−^* interactions in eye discs show small *scrib^−^* clones identified by loss of Scrib expression (red in **A**, grey in **A’**) and adjacent *Ras^V12^* clones (GFP positive green in **A**, grey in **A”**). (**B**–**B”**) The YZ projection of the yellow line in **A** shows a multi-layered *Ras^V12^* clone (GFP). The red asterisks mark the *scrib^−^* clones (GFP negative in **B**,**B”**), the yellow arrowhead marks the disc proper, and the blue arrowhead marks the peripodial membrane. (**C**–**F**) Eye discs show expression of Casp3*(**C** red, **C’** grey), Wg (**D** red, **D’** grey), p-JNK (**E** red, **E’** grey), and Yki (**F** red, **F’** grey), in *Ras^V12^//scrib^−^* interclonal interactions. Note upregulation of Casp3*, Wg and Yki at the boundary of *Ras^V12^* clones. Scale bar = 25 μm.

**Figure 7 cancers-16-01768-f007:**
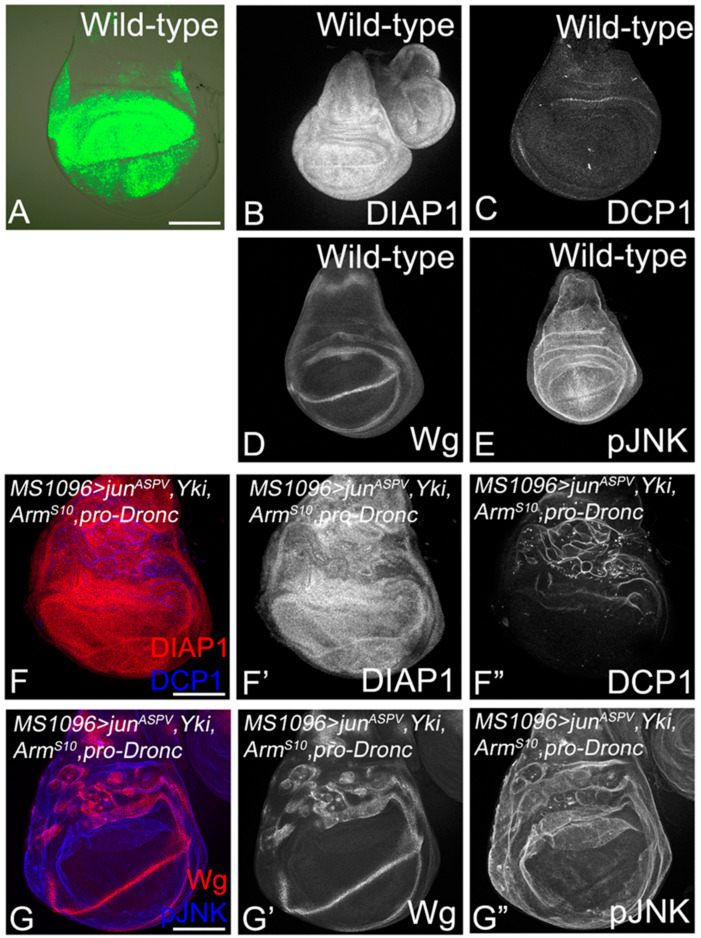
Loss of polarity is required for the Wg–Dronc–JNK–Yki network to drive tumor growth. (**A**–**E**) Control wing imaginal discs from (**A**) *MS1096 Gal4 > UASGFP*, and wild-type wing discs stained for (**B**) DIAP1, (**C**) DCP1, (**D**) Wg, and (**E**) pJNK are shown. (**F**–**G”**) Panels show wing imaginal discs from *MS1096 Gal4 > UASjun^aspv^*, *UASYki*, *UASproDronc*, *UASArm^S10^* stained for expression of DIAP1 ((**F**) red, (**F’**) grey), DCP1 ((**F**) blue, (**F”**) grey), Wg ((**G**) red, (**G’**) grey), and pJNK ((**G**) blue, (**G”**) grey). Disc magnification and orientation are identical in all panels. Scale bar = 25 µm.

## Data Availability

Data is maintained within this article.

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
