# Peer review of "A Tumor-Specific Molecular Network Promotes Tumor Growth in Drosophila by Enforcing a Jun N-Terminal Kinase–Yorkie Feedforward Loop"

_cancers, 2024, doi:10.3390/cancers16091768_

Round 1
Reviewer 1 Report (Previous Reviewer 1)
Comments and Suggestions for Authors Waghmare and colleagues have addressed how cancer cells re-wire their regulatory networks to become over-proliferative and malignant. They have produced a good body of work to support their proposal that, in Drosophila, oncogenic-transformed cells that lose polarity produce cell-autonomously the Wg signalling molecule, which activates the caspase Dronc (in a non-apoptotic function), and both the JNK and Hippo pathways to initiate a self-sustained stimulation of growth, invasion and survival. This is a new submission of a manuscript (cancers #2734242) that I reviewed previously. Back then I thought the work merited publication with a few modifications, and the body of work is essentially the same, therefore I find no new areas to highlight for improvement (and I feel it would not be fair to do so). Therefore, I will go through the issues I highlighted initially. Major point #1 - Wg signalling epistasis and readout analysis- Epistasis analyses using UAS-sgg[S9A] - is it a proper Wg signalling inhibitor? The authors replied satisfactorily to this in their revised version of ms cancers-2734242.
- Can the inhibition of the Wg pathway be confirmed using anti-Wg as a readout in RasV12, scrib– cells? The authors have cited 3 papers where it is shown that Wg is its own target. However, these papers all refer to the wild-type embryonic epidermis and not the RasV12, scrib– cancer model in the imaginal wing disc. I remain unconvinced. There are, however, other readouts that the authors could use, like the negative feedback factors notum (10.1101/gad.991802) or frizzled3 (10.1242/dev.126.20.4421; 10.1016/S0925-4773(99)00313-5)
- Description of image acquisition conditions, pixel selection for fluorescence intensity determination, and normalization approach. The authors do this now, though with minimal detail. I am afraid I still have a few unclarified issues, though. Figures 1H and K seem to have escaped the normalisation: their y-axis should show the wild type average value at around 1 across images. However, in H this average is around 50, and in K around 80 or 90. These do not look like normalized values. Moreover, you have a value over 300 in 1K, which is not consistent either with using directly the "Arbitrary Intensity" of the pixel intensity value (because it is out of scale in an 8-bit image, and it would be near black in a 16-bit image). More details are needed here, sorry. Also, most wild-type bar plots have no error bars - it seems difficult to understand how individual measurements of fluorescence intensity would have no variability whatsoever in the wild-type.
- Statistics. These are more explicit now, but the authors still use t-test without justification - they must be preceded by a normality test, or an alternative test should be used.
- Presenting bar plots with error bars: this is done now.
- Using pixel number as a measure of area instead of absolute measurements - this has not been addressed (though it was a minor point)
- Panels in figure 4 A-D are confusing as the direction in which you follow the order changes between A' and B, and between B' and C. Also, the blue panel in A should be shown separately. This has not been addressed.
- Quantification in figure 4M does not clearly correspond to what I see in the images K, K'. This has not been addressed as far as I could see.
- I cannot see downregulation of Yki in fig S5CD as claimed by authors. Some clones may show this effect, but many do not seem to have this response. This has not been addressed.
- Statement in lines 392-4: "Thus, loss of scrib had clear and distinguishable effects from Yki overexpression (Fig. 6A) on the four signals." [My highlight] I am not sure what is the point of comparison to the results from figure 5 and S5 that allow the authors to make this claim. This has not been addressed as far as I could see.
- The authors claim (line 477) that Wg is upregulated Fig S6H, but I remain unconvinced - this is only seen in part of the prospective hinge, and not in the wing blade nor even the wing margin.
Author Response
Please see the attachment

Reviewer 2 Report (New Reviewer)
Comments and Suggestions for Authors
This work by Waghmare et al. has used a Drosophila in vivo tumor model to investigate mechanisms of tumor development. Specifically, they identified a molecular network involving Wg, Dronc, JNK, and Yki signaling pathways that are required for the growth of Ras-V12, scrib- tumor cells. With Wg acting upstream of Dronc, both Wg and Dronc converge onto a JNK-Yki positive feedback loop. Although the concept of oncogenic cooperation has been well established through cancer research over the past decades, this work has revealed how some of the oncogenic pathways are functionally related to promote tumorigenesis. Some minor concerns are listed below:
1) Line 46, the graphical abstract, the apical-basal polarity needs to be marked/highlighted within epithelial WT cells.
2) Figure 1, D & E panels, for a better qualitative organ size comparison of the two genotypes, low magnification images are needed to show the entire eye-antennal discs either in Figure 1 or the supplemental figure.
3) Figure 2C, the middle-bottom panel, the label of "Casp3" is missing.
4) Lines 368-369, "...RasV12,scrib-cells in combinations where either Yki (Fig. 4A-D) ... are downregulated". This description doesn't appear to be consistent with the bar representation shown in Figure 4D, the middle Yki panel.
5) Line 441, "(Fig. 6C, C’)" to be revised as (Fig. 6D, D'); Line 442, "(Fig. 6D, D’)" to be revised as (Fig. 6E, E').
6) For the Figure 6F, F', the Yki results need to be described.
7) Lines 568-569, "... steep differences in the non-cell autonomous spread of Yki activity between scrib- (Fig. 5), RasV12 scrib- (Fig. S1C,C’) ...", it is inconsistent with what are actually shown in Figure S1C,C' about the Casp3* levels.
8) The entire Figure S1 needs to be presented.
Author Response
Please see the attachment

Reviewer 3 Report (New Reviewer)
Comments and Suggestions for Authors
The manuscript by Waghmare et al., entitled "A Tumour-Specific Molecular Network Promotes Tumour Growth in Drosophila by enforcing a JNK-YKI Feedforward Loop" addresses the mechanisms that control the growth behavior of RASV12/scrib- mutant clones in the developing imaginal discs of Drosophila.
The authors have tackled this question by carrying out various epistatic genetic experiments which have shown that the activity of the Wingless (Wg), Dronc, Yorkie (Yki) and JNK pathways is required to sustain the growth of RASV12/scrib- mutant cells.
While the general interest of the article is obvious, some of the conclusions reached by the authors need to be strengthened.
General comments
1)- The quality of the main figures presented in the pdf is very poor, and as most of the conclusions are based on immunostaining, it is not always easy to follow the interpretation made by the authors. In addition, most experiments were performed in the eye disc, where cell differentiation after the furrow makes staining and analysis more difficult to believe. On the contrary, the specific and characteristic expression pattern of Wg in the wing disc facilitates interpretation of the results obtained in this tissue (see Figure 7). For example, in Figure 1, the authors say, and I know this is correct, that Dronc is ubiquitously expressed in the wild-type imaginal disc. But probably due to folding, Dronc staining appears higher in the lower part of the disc. In Figure 1C, the authors claim that DIAP1 is down-regulated, but this is not convincing from the image they show. There are other examples like this throughout the manuscript (DIAP1-LacZ in Yki gain-of-function in FigS1, DIAP1 staining in Sup 3C...).
2)- More importantly, one of the main conclusions of the article, that Wg somehow acts upstream of Dronc, JNK and Yki, based on the data presented in Fig 3, is unconvincing. Firstly, Wg is not upregulated in all clones and is sometimes upregulated in regions devoid of clones (Fig 3C and D). Once again, further analysis in the wing disc could be useful, as this point is very important for the conclusion of the article. In addition, the authors show that Yki gain-of-function is sufficient to induce ectopic Wg expression (Fig S6). This should be discussed further.
3)- I don't know how the authors were able to identify the A-P boundary in their wing disc experiments without any markers to look at (Figure 5).
4)- I'm not sure that the title clearly highlights the subject of the article. YKI should be written Yorkie both in the title and in the first quote of the article. I don't think this is the case.
Minor comments
1)- The standard deviation does not appear in figure Sup2 if the experiments were carried out three times as indicated in the method section.
2)- The lettering is not uniform along the entire length of the paper. Letters sometimes appear at the top or bottom of the figure.
3)- I am surprised to see a non-autonomous induction of cell death by Yki depletion. This should be discussed further.
4)- It would have been interesting to have the appropriate control (RaSV12/scrib-) in Figure 2, Figure 3, Figure 4.
5)- In the text, I'm not sure that the reference to Figure 7 is accurate (e.g. "figure 7B" didn't exist).
Round 2
Reviewer 3 Report (New Reviewer)
Comments and Suggestions for Authors
The authors have taken my comments into account.
Sounds good to me.
This manuscript is a resubmission of an earlier submission. The following is a list of the peer review reports and author responses from that submission.
Round 1
Reviewer 1 Report
Comments and Suggestions for Authors Waghmare and colleagues have addressed how cancer cells re-wire their regulatory networks to become over-proliferative and malignant. They have produced a good body of work to support their proposal that, in Drosophila, oncogenic-transformed cells that lose polarity produce cell-autonomously the Wg signalling molecule, which activates the caspase Dronc (in a non-apoptotic function), and both the JNK and Hippo pathways to initiate a self-sustained stimulation of growth, invasion and survival. I believe this work merits publication but I have a few objections to the presentation of some of the data as well as the interpretation of some of the results. I am not sure that the study requires significant additional experimentation necessarily, but the manuscript would certainly require some modifications to address the issues I raise below. The most important thing that I think needs further clarification are the epistasis analyses using UAS-sgg[S9A]. Shaggy (GSK3beta) acts downstream the pathway ligand to reduce the concentration and activation of the nuclear effector, Armadillo. However, Sgg is a complex kinase with multiple phosphorylatable residues itself, so its regulation is not so straight-forward. Papadopoulou et al., 2004 (10.1128/MCB.24.11.4909-4919.2004) show that the S9A substitution does not inhibit Wg signalling. Moreover, even if it did, it is not clear to me at all how the inhibition of the pathway can be confirmed "with anti-Wg antibodies" (line 304), unless there is a well-established, cell-autonomous, positive feedback loop between Wg signalling and wg expression, described to function in RasV12, scrib– cells. Without a clear rationale for using this transgene, I find it difficult to accept the authors' claims they make about the role of Wg signalling. It also makes it difficult to interpret the epistasis analyses with Dronc, JNK and Yki. The other issue that is absolutely essential to clarify is how the quantitative analyses of the images were done, as well as the statistics. I think that just comparing mean pixel values from different images, with no reference to the image acquisition conditions, is incorrect. I would also expect that the analysis description goes to some lengths into convincing me that they have indeed acquired the images from different speciments using the same imaging conditions, ideally in the same session or with at least 30 minutes of laser warming up before acquisition (unless the laser sources do not need warm-up), and that the samples were prepared with the same batch of antibody solution for both primary and secondary. When these conditions cannot be met (which is most of the times, for practical reasons), I would expect some internal normalization approach, especially when clones are used, so the signal from wild-type areas far from the clones are used as a normalization factor for the signal within the clones. Besides, t-tests are only acceptable if the data has passed first a normality test, and representations of normalized pixel intensity and clonal size should have standard deviation or standard error bars (if they have to be bar plots at all). Finally, pixel number as a measure of area is very easily turned into absolute measurements using the metadata of the images - this is a minor point but it would improve the presentation of the data. Minor points:- The fact that caspases have non-apoptotic roles is well known and is starting to be characterised in detail (see, for instance, the work of LA Baena-López), so it is a bit unnecessary to repeat over and over that your finding of Dronc being upregulated in growing tumours is 'paradoxical'. Rather, the work of the people who have shown that this is the not the case should be cited.
- Some transgenes are not properly described: At least UAS-dronc[RNAi], UAS-Yki[N+CRNAi] should be identified with the appropriate reference and/or collection number.
- The authors use 'down/upregulated' a bit indiscriminately, to either refer to experimental knock-down, changes in gene expression, or inactivation of a signaling pathway (e.g. line 256). I think the readability of the text would improve if these uses were more precise.
- Panels in figure 4 A-D are confusing as the direction in which you follow the order changes between A' and B, and between B' and C. the blue panel in A should be shown separately.
- Quantification in figure 4M does not clearly correspond to what I see in the images K, K'.
- I cannot see downregulation of Yki in fig S5CD as claimed by authors.
- Statement in lines 392-4: "Thus, loss of scrib had clear and distinguishable effects from Yki overexpression (Fig. 6A) on the four signals." [My highlight] I am not sure what is the point of comparison to the results from figure 5 and S5 that allow the authors to make this claim.
- Lines 394-5: "We found that scrib- cells grew to significantly larger sizes in the posterior (P) compartment..." Do the authors mean cells or clones?
- The authors claim that the self-amplification loop between JNK and Yki is specific from cells that lose polarity, but in figure 5D I can see a scrib– anterior clone that looks like it is upregulating pJNK.
- In figure 5AB the blue panel (ex-lacZ) needs to be shown separately, and the label in the panels should read ex-lacZ and not betaGal, so the readers can interpret it at a glance.
- In Figure 7, the wild-type disk should be shown with the expression of the markers.
- Line 455-6: "overexpression of Yki (MS1096>Yki) is capable of driving hyperplasia by upregulation of DIAP1, Wg and pJNK" [my highlight]. "By" should be "and" as the authors do not do an epistatic analysis at this point.
- I cannot see any upregulation of Wg in Fig S6G-H as the authors claim.
- In the discussion, you claim to "have found that JNK, Yki, Dronc, and Wg were all required for aggressive growth of RasV12,scrib- induced tumours" (lines 524-5). However the role of JNK and Hippo pathways in these clones had been described a while ago in Igaki et al., 2006 (10.1016/j.cub.2006.04.042) and in Doggett et al., 2011 (10.1186/1471-213X-11-11). [Those are the references I could find. There may be others.] Igaki et al in particular is cited in the manuscript as ref 30 but it is not cited in the context of this finding. This should be corrected.
- In lines 534-6 the authors claim to have found that "upregulation of JNK and Yki in normal epithelial cells with intact polarity is not sufficient to induce this network and tumour growth". I am not convinced this is a new discovery, either. For instance, Ma et al., 2014 (10.1073/pnas.1415020112) show that activation of Yki leads to activation of JNK, so all existing literature showing that Yki activation does not lead to malignant growths per se could make the same claim. In addition, in a different context Liu et al., 2016 (10.1038/srep38003) show that Yki and JNK cooperate to make folds in the wing imaginal disk. One could argue that the interpretation of the authors is novel, and I could be on board with that, but there should a reference to previous work that have observed co-activation of JNK/Yki in polarised imaginal disk cells without malignant tumours.
- The authors could also back-up this last claim in a more conclusive manner if they did the simultaneous overexpression of UAS-jun[aspv], UAS-Yki, UAS-proDronc, and UAS-arm[S10] using the MS1096 driver, also with a UAS-scrib[RNAi] transgene. I appreciate this may be a bit challenging but I presume it is doable.
- A final thought is the fact that the conclusions are pieced together using information from both the wing and the eye disks. It is difficult to follow which parts have been done in parallel or only in one system. The discussion should highlight this.
Author Response
For research article:
Manuscript ID: cancers-2734242
Title: A Tumour-Specific Molecular Network Promotes Tumour Growth in Drosophila by Enforcing a JNK-YKI Feedforward Loop
|
Response to Reviewer 1 Comments
|
||
|
1. Summary |
|
|
|
Thank you very much for taking the time to review this manuscript. Please find the detailed responses below and the corresponding revisions in red font in the re-submitted files.
|
||
|
2. Questions for General Evaluation |
Reviewer’s Evaluation |
Response and Revisions |
|
Does the introduction provide sufficient background and include all relevant references? |
Yes |
Thank you |
|
Are all the cited references relevant to the research? |
Must be improved |
Thanks, we have added citations in response to the reviewer’s comments |
|
Is the research design appropriate? |
Can be improved |
Improved statistical analyses and details have been added in the revised manuscript. |
|
Are the methods adequately described? |
Must be improved |
We have added information to make the methods clear and accessible. |
|
Are the results clearly presented? |
Must be improved |
We have revised and edited the results section to clarify our findings. |
|
Are the conclusions supported by the results? |
Can be improved |
Following the reviewer’s suggestions, the manuscript has been revised. |
|
3. Point-by-point response to Comments and Suggestions for Authors |
||
|
Comments 1: Waghmare and colleagues have addressed how cancer cells re-wire their regulatory networks to become over-proliferative and malignant. They have produced a good body of work to support their proposal that, in Drosophila, oncogenic-transformed cells that lose polarity produce cell-autonomously the Wg signalling molecule, which activates the caspase Dronc (in a non-apoptotic function), and both the JNK and Hippo pathways to initiate a self-sustained stimulation of growth, invasion and survival. I believe this work merits publication but I have a few objections to the presentation of some of the data as well as the interpretation of some of the results. I am not sure that the study requires significant additional experimentation necessarily, but the manuscript would certainly require some modifications to address the issues I raise below. The most important thing that I think needs further clarification are the epistasis analyses using UAS-sgg[S9A]. Shaggy (GSK3beta) acts downstream the pathway ligand to reduce the concentration and activation of the nuclear effector, Armadillo. However, Sgg is a complex kinase with multiple phosphorylatable residues itself, so its regulation is not so straight-forward. Papadopoulou et al., 2004 (10.1128/MCB.24.11.4909-4919.2004) show that the S9A substitution does not inhibit Wg signalling. Moreover, even if it did, it is not clear to me at all how the inhibition of the pathway can be confirmed "with anti-Wg antibodies" (line 304), unless there is a well-established, cell-autonomous, positive feedback loop between Wg signalling and wg expression, described to function in RasV12, scrib– cells. Without a clear rationale for using this transgene, I find it difficult to accept the authors' claims they make about the role of Wg signalling. It also makes it difficult to interpret the epistasis analyses with Dronc, JNK and Yki.
|
||
|
Response 1: Thank you for pointing this out. I/We agree with this comment. The reviewer has raised two important points: (a) is the use of UAS-sggS9A (BL#5255) transgene appropriate in the context of Wg inhibition, and (b) is there evidence for Wg expression being regulated by Wg signaling via a feedback loop to justify the use of Wg antibodies for checking inhibition of the Wg pathway? (a) UAS-SggS9A is a well-established transgene that dominantly blocks Wg signaling (Hazlett et al., 1998). This transgene has been validated and previously used and published by several labs (including our own). The reviewer points to the complexities of Wg regulation as shown by Papadopoulou et al., 2004; however, our intent is to block canonical WG signaling which is a well-known function of this transgene. (b) Wg is a target of its own signaling (Ingham and Hidalgo, 1993; Manoukian et al., 1995; Yoffe et al., 1995). Thus, in the eye, inhibition of Wg pathway leads to suppression of Wg expression. Therefore, we made UAS SggS9A control clones (Fig. S3B-B”) and confirmed that endogenous expression of Wg was suppressed in UASSggS9A clones (shown by yellow arrows in Fig. S3B”).
The reviewer also states that ‘Sgg is a complex kinase with multiple phosphorylatable residues itself, so its regulation is not so straight-forward.’ We agree with the reviewer’s comment and have been testing other Wg inhibitors for our studies. We have found that impairing Wg in the Wg-producing cells using RNAi to knockdown porcupine (UAS-porcRNAi) in RasV12 scrib- – clones phenocopies the effects of expression of UAS-SggS9A in RasV12, scrib- clones. Recently, we have also confirmed that co-expression of dominant negative form of TCF (UAS-TCFDN) in RasV12, scrib- clones also phenocopies the effects of expression of UAS-SggS9A. .Given the similarity of the phenotypes, we went ahead with further characterizing the effects of UAS-SggS9A for our studies.
We have added the following to clarify the use of UAS-SggS9A and the Wg antibody expression in the results section: Line 270-275: “---we tested their requirement by downregulating these signals in RasV12,scrib- clones (Fig.2) using well-established transgenes, for example., RNAi to knockdown Dronc which impairs Caspase mediated signaling [UAS-DroncRNAi][50], the dominant-negative form of Bsk which is a potent suppressor of JNK signaling [UAS-BskDN][37], UAS-SggS9A which dominantly blocks Wg signaling [52] and UAS-YkiN+CRNAi which causes inactivation of Yki-mediated signalling [34].”
Line 316-318 “Wg is a target of its own signaling [54–56]. Therefore, we made UAS-SggS9A control clones (Fig. S3B-B”) and confirmed that endogenous expression of Wg was suppressed in UAS-SggS9A clones using anti-Wg antibody (shown by yellow arrow in Fig. S3B”).”
|
||
|
Comments 2: The other issue that is absolutely essential to clarify is how the quantitative analyses of the images were done, as well as the statistics. I think that just comparing mean pixel values from different images, with no reference to the image acquisition conditions, is incorrect. I would also expect that the analysis description goes to some lengths into convincing me that they have indeed acquired the images from different speciments using the same imaging conditions, ideally in the same session or with at least 30 minutes of laser warming up before acquisition (unless the laser sources do not need warm-up), and that the samples were prepared with the same batch of antibody solution for both primary and secondary. When these conditions cannot be met (which is most of the times, for practical reasons), I would expect some internal normalization approach, especially when clones are used, so the signal from wild-type areas far from the clones are used as a normalization factor for the signal within the clones. Besides, t-tests are only acceptable if the data has passed first a normality test, and representations of normalized pixel intensity and clonal size should have standard deviation or standard error bars (if they have to be bar plots at all). Finally, pixel number as a measure of area is very easily turned into absolute measurements using the metadata of the images - this is a minor point but it would improve the presentation of the data.
|
||
|
Response 2: Thanks for pointing this out. We have described the methods and accordingly modified the figures to show the updated quantification. Section Materials and Methods, section 2.4, Lines 152-163” “Statistical analyses were performed using Microsoft Excel 2013. The magnetic lasso tool was used for clone size comparison (Fig. 1, Wild type n=11, rest n=25). Mean pixel values of clone area were obtained using the Histogram function in Photoshop CS6, and analysed using a two-tailed Student’s t-test assuming statistical significance at p<0.05. Intensity plots (Fig. 1, S1), were made using the plot profile function in ImageJ to find changes in pixel intensity. Dot plots were generated by calculating average signal intensity both inside (GFP-positive) and outside (GFP-negative) the clone in the Dronc or Wg channels (Fig. 1, S1) (n=5). The average wild-type values were used to find the normalization factor to determine and calculate differences in intensity levels between wild-type, scrib-, and RasV12; scrib- clones. Similarly, the normalization factor was calculated using wild-type values (GFP-negative areas) to assess changes in expression of Wg (Fig. 3), Yki, pJNK, and Dronc (Fig.4). All graphs were plotted using GraphPad Prism8.0”
Minor points: i. The fact that caspases have non-apoptotic roles is well known and is starting to be characterised in detail (see, for instance, the work of LA Baena-López), so it is a bit unnecessary to repeat over and over that your finding of Dronc being upregulated in growing tumours is 'paradoxical'. Rather, the work of the people who have shown that this is the not the case should be cited. Response: Thanks, we have added citations to include the apoptotic and nonapoptotic roles of Dronc in the context of cancer, and toned down the ‘paradoxical role’. ii. Some transgenes are not properly described: At least UAS-dronc[RNAi], UAS-Yki[N+CRNAi] should be identified with the appropriate reference and/or collection number. Response: Thanks, we have added the citations and collection numbers for all transgenes used in this study. See revised Materials and Methods section 2.1 (Lines 106-114) for details.
iii. The authors use 'down/upregulated' a bit indiscriminately, to either refer to experimental knock-down, changes in gene expression, or inactivation of a signaling pathway (e.g. line 256). I think the readability of the text would improve if these uses were more precise. Response: Thanks, we have carefully edited the manuscript to more precisely convey the context for down/upregulated.
iv. Panels in figure 4 A-D are confusing as the direction in which you follow the order changes between A' and B, and between B' and C. the blue panel in A should be shown separately. Response: Thanks, we have made the suggested change in revised Fig. 4 v. Quantification in figure 4M does not clearly correspond to what I see in the images K, K'. Response: See revised Fig 4M, the levels of Dronc and pJNK are reduced when UAS-SggS9A is co-expressed in the RasV12, scrib- clones.
vi. I cannot see downregulation of Yki in fig S5CD as claimed by authors. Response: Fig S5C shows arrowheads where diap1-lacZ is downregulated, suggesting reduced Yki activity. In Fig S5D, clear suppression of Yki is not visible in the disc proper for clones. In response to the reviewer’s comment, we have revised the manuscript to: Lines 400-401: “-- scrib- clones upregulated JNK (Fig. S5A-B’), downregulated Yki activity as seen by suppression of diap1-lacZ (Fig. S5C-D’ - -.” vii. Statement in lines 392-4: "Thus, loss of scrib had clear and distinguishable effects from Yki overexpression (Fig. 6A) on the four signals." [My highlight] I am not sure what is the point of comparison to the results from figure 5 and S5 that allow the authors to make this claim. Response: Yki overexpression causes upregulation of Diap1 and other markers, and results in growth, whereas loss of scrib (Fig. S5) causes downregulation of Diap1 and elimination of clones due to cell competition. These are clear and distinguishable effects, that we were referring to. To clarify our statement, we have edited the text preceding this statement as follows in Lines 403-405 in the revised manuscript: “Consistent with published data, over-expression of Yki resulted in robust overgrowth due induction of Yki activity leading to upregulation of JNK[14,16], activation of DIAP1 (Fig. S1F), ectopic induction of Wg [16] and suppression of Dronc [50]. Thus, loss of scrib had clear and distinguishable effects ---" viii. Lines 394-5: "We found that scrib- cells grew to significantly larger sizes in the posterior (P) compartment..." Do the authors mean cells or clones? Response: We mean clones, we have made the edit in the revised manuscript, see Line 408 in the revised manuscript. ix. The authors claim that the self-amplification loop between JNK and Yki is specific from cells that lose polarity, but in figure 5D I can see a scrib– anterior clone that looks like it is upregulating pJNK. Response: Thanks for your comment. In the anterior compartment, the scrib2- clones are in wild-type background and so are subject to JNK mediated cell competition and elimination (Igaki et al., 2006). Kindly note that in the anterior compartment very few and small scrib- clones are recovered. pJNK is induced in these A compartment specific scrib- clones, but Yki is not (Fig. 5). Thus, these A compartment specific clones do not establish the self-amplification loop. x. In figure 5AB the blue panel (ex-lacZ) needs to be shown separately, and the label in the panels should read ex-lacZ and not betaGal, so the readers can interpret it at a glance. Response: Thanks for this excellent suggestion. In Fig 5B, we have shown the ex-lacZ channel separately, and changed the labelling from beta-gal to ex-lacZ as suggested by the reviewer. xi. In Figure 7, the wild-type disk should be shown with the expression of the markers. Response: Thanks, we have added the wild-type images in the revised manuscript. xii. Line 455-6: "overexpression of Yki (MS1096>Yki) is capable of driving hyperplasia by upregulation of DIAP1, Wg and pJNK" [my highlight]. "By" should be "and" as the authors do not do an epistatic analysis at this point. Response: Thanks, we have made the suggested change, see revised manuscript Line 470. xiii. I cannot see any upregulation of Wg in Fig S6G-H as the authors claim. Response: Thanks for your comment. We have added arrowheads in wing pouch region to show the increase in the thickness of the Wg stripe (Fig. S6) which represents its upregulation, when compared to wild type (Fig. 7D). xiv. In the discussion, you claim to "have found that JNK, Yki, Dronc, and Wg were all required for aggressive growth of RasV12,scrib- induced tumours" (lines 524-5). However the role of JNK and Hippo pathways in these clones had been described a while ago in Igaki et al., 2006 (10.1016/j.cub.2006.04.042) and in Doggett et al., 2011 (10.1186/1471-213X-11-11). [Those are the references I could find. There may be others.] Igaki et al in particular is cited in the manuscript as ref 30 but it is not cited in the context of this finding. This should be corrected. Response: Thanks for your comment. We have added these and other references in response to your comments. xv. In lines 534-6 the authors claim to have found that "upregulation of JNK and Yki in normal epithelial cells with intact polarity is not sufficient to induce this network and tumour growth". I am not convinced this is a new discovery, either. For instance, Ma et al., 2014 (10.1073/pnas.1415020112) show that activation of Yki leads to activation of JNK, so all existing literature showing that Yki activation does not lead to malignant growths per se could make the same claim. In addition, in a different context Liu et al., 2016 (10.1038/srep38003) show that Yki and JNK cooperate to make folds in the wing imaginal disk. One could argue that the interpretation of the authors is novel, and I could be on board with that, but there should a reference to previous work that have observed co-activation of JNK/Yki in polarised imaginal disk cells without malignant tumours. Response: Thanks, we have referenced these works in the discussion. See Lines 555-556 “which is consistent with other instances of developmental interactions between Yki and JNK in polarized epithelia [78, 79].” xvi. The authors could also back-up this last claim in a more conclusive manner if they did the simultaneous overexpression of UAS-jun[aspv], UAS-Yki, UAS-proDronc, and UAS-arm[S10] using the MS1096 driver, also with a UAS-scrib[RNAi] transgene. I appreciate this may be a bit challenging but I presume it is doable. Response: This is an excellent suggestion; however, we faced technical challenges while trying to do this earlier. We made MS1096 Gal4; UAs junASPV flies and crossed them to UASArmS10; UASYki; UASproDronc flies for this experiment (data presented in the manuscript). We also tried to generate MS1096Gal4; UASjunASPV; UASscribRNAi for the experiment you suggested. However, this line showed very high pupal lethality. The few escapers we got were not healthy and died within a day or two of eclosion. We were not successful in completing this experiment using identical conditions (25oC) as the experiments described in Fig.7. xvii. A final thought is the fact that the conclusions are pieced together using information from both the wing and the eye disks. It is difficult to follow which parts have been done in parallel or only in one system. The discussion should highlight this. Response: Thanks for your comment. Most of the data is from work done on the eye discs. To clarify, we have added the following to the discussion, lines 506-510: “Here we present evidence to show that a tumour cell specific network is formed in RasV12,scrib-/- tumours in eye imaginal discs. This network is comprised of Yki, JNK, Wg and Caspases that acts to promote fitness and aggressive growth (Fig. 1, S1). Our findings from the eye discs are relevant to other epithelia like the wing discs as the tumour specific network is recapitulated in the en>Yki; scrib- wing tumours (Fig. 5).”
|
||
|
4. Response to Comments on the Quality of English Language |
||
|
Point 1: (x) English language fine. No issues detected |
||
|
Response 1: Thank you.
|
||
|
5. Additional clarifications |
||
|
[Here, mention any other clarifications you would like to provide to the journal editor/reviewer.] |
||

Reviewer 2 Report
Comments and Suggestions for Authors
Cancer cells rapidly expand through altered signaling interactions. The manuscript by Waghmare et al presents exciting findings about the cooperation of Wg, Dronc, JNK and Yki in the growth of RasV12, scrib-tumor using Drosophila as a model. However, before publication, the following issues need to be addressed:
1. For quantification of upregulation of Dronc in wild type, Scrib, Rasv12scrib clones (Fig 1H), pixel intensity should be normalized with neighboring cells outside the clone. This will take care of sample-to-sample variation of Dronc staining signal. Note the huge difference in Dronc staining signal in cells outside the clone in Fig 1F and Fig 1G.
2. Quantification is required for data in Fig 1M & O.
3. At a few places, the authors are not differentiating what is published and what is novel. For instance, in line 240: the claim is Yki, JNK, Wg and Caspases are upregulated. Out of these, Yki and JNK are previously shown to be upregulated in scrib mutant cells. Citations are required for those claims.
4. Number of samples examined is missing in Fig 2F. N should be included for all the quantifications, as in the graph of Fig 3.
5. In the current version, it will be difficult for people outside Drosophila to appreciate the results.
Comments on the Quality of English LanguageEnglish grammar needs to be taken care of in the manuscript.
Author Response
For research article
Manuscript ID: cancers-2734242
Title: A Tumour-Specific Molecular Network Promotes Tumour Growth in Drosophila by Enforcing a JNK-YKI Feedforward Loop
|
Response to Reviewer 2 Comments
|
||
|
1. Summary |
|
|
|
Thank you very much for taking the time to review this manuscript. Please find the detailed responses below and the corresponding revisions in red font in the re-submitted files.
|
||
|
2. Questions for General Evaluation |
Reviewer’s Evaluation |
Response and Revisions |
|
Does the introduction provide sufficient background and include all relevant references? |
Yes |
Thanks |
|
Are all the cited references relevant to the research? |
Yes |
Thanks |
|
Is the research design appropriate? |
Yes |
Thanks |
|
Are the methods adequately described? |
Yes |
Thanks |
|
Are the results clearly presented? |
Can be improved |
We have revised our manuscript to clarify the results |
|
Are the conclusions supported by the results?
|
Can be improved |
Following the reviewer’s suggestions, the manuscript has been revised. |
|
3. Point-by-point response to Comments and Suggestions for Authors |
||
|
Comments 1: For quantification of upregulation of Dronc in wild type, Scrib, Rasv12scrib clones (Fig 1H), pixel intensity should be normalized with neighboring cells outside the clone. This will take care of sample-to-sample variation of Dronc staining signal. Note the huge difference in Dronc staining signal in cells outside the clone in Fig 1F and Fig 1G. |
||
|
Response 1: Thank you for pointing this out. We agree with this comment. Therefore, we have normalized the pixel intensity as suggested by the reviewer in the revised manuscript. See revised Fig. 1, S1 in this context. All editorial changes in the revised manuscript are in red.
|
||
|
Comments 2: Quantification is required for data in Fig 1M & O. |
||
|
Response 2: Agree. We have added this data to revised figure 1 (Fig. 1N, Q) and in the results section of the revised manuscript.
Comments 3: At a few places, the authors are not differentiating what is published and what is novel. For instance, in line 240: the claim is Yki, JNK, Wg and Caspases are upregulated. Out of these, Yki and JNK are previously shown to be upregulated in scrib mutant cells. Citations are required for those claims. Response 3: Thanks for pointing this out, we have added citations in the revised manuscript.
Comments 4: Number of samples examined is missing in Fig 2F. N should be included for all the quantifications, as in the graph of Fig 3. Response 4: Thanks, we have added this information in the revised manuscript.
Comments 5: In the current version, it will be difficult for people outside Drosophila to appreciate the results. Response 5: We have tried to reduce jargon to make this manuscript accessible to non-Drosophila readership of the journal.
|
||
|
4. Response to Comments on the Quality of English Language |
||
|
Point 1: English grammar needs to be taken care of in the manuscript. |
||
|
Response 1: The manuscript has been edited by a native English speaker.
|
||
|
5. Additional clarifications |
||
|
|
||

Reviewer 3 Report
Comments and Suggestions for Authors
In this MS, the authors use Drosophila to analyse a genetic network controlling tumor growth. The authors use a classic tumor model, the Ras-scrib model. The authors analyse different genes in those tutors and claim to have identified a gene network driving tumorigenesis. However, the manuscript lacks cohesion. More importantly, the results are in most of the cases over interpreted and the conclusions are not supported by the results presented. many parts of the work lack quantifications and are supported by single examples. These flaws can be illustrated in the following comments pertaining to the first part of the manuscript. Similar flows are repetitively present in he rest of the manuscript.
See those specific comments below:
Panels do not have scale bars. This element is crucial when analyzing processes affecting clone and tissue size.
Fig 1C. What part of the disc is that? The authors should have used a general marker as DAPI or phalloidin to show the shape of the disc.
What is shown in 1E? I only see some green patches but the shape of the disc can´t be observed.
”however the discs remain monolayered and clones did not form tumours (Fig. S1B)” where is it shown that the clones are monolayered? I can´t find any panel showing that.
“In comparison to wild-type clones (Fig. 1B-B’), in RasV12,scrib- clones a bulk of cell death was induced in the wild-type cells surrounding the clone (Fig. 1C, C’).” This is partially true. There is some clear caspase activity in GFP positive cells close to the arrowhead at the bottom.
“Similarly in the wing discs, compared to wild-type (Fig.1D) RasV12,scrib- 187 clones grew to large invasive tumours (Fig. 1E).” I can´t see any invasion in that panel.
The authors refer recurrently to Ay-Gal4. Is that correct or do they mean eyeless-gal4 (ey-Gal4).
“Increased Yki activity due to Yki overexpression in ‘flp-out’ (AyGAL4>Yki) clones showed robust induction of diap1-lacZ (Fig. S1D).” It´s impossible to see Diap1 expression in FS1. The GFP levels are masking the red signal.
The authors claim tha Dronc is induced in Ras-scrib clones, F1G. But Dronc seems to be upregulated in a patch of cells with a similar shape of that clone in wild type discs (F1F&F´).
Wg is claimed to be induced in those clones. However, it is upregulated in some cells of the clone (Fig1J&J´). How do the authors reconcile that with their model? The same coment applies for Dronc-lacZ (F1M).
Comments on the Quality of English LanguageIt can be improved.
Author Response
For research article
Manuscript ID: cancers-2734242
Title: A Tumour-Specific Molecular Network Promotes Tumour Growth in Drosophila by Enforcing a JNK-YKI Feedforward Loop
|
Response to Reviewer 3 Comments
|
|
|||
|
1. Summary |
|
|
|
|
|
Thank you very much for taking the time to review this manuscript. Please find the detailed responses below and the corresponding revisions in track changes in the re-submitted files.
|
|
|||
|
2. Questions for General Evaluation |
Reviewer’s Evaluation |
Response and Revisions |
|
|
|
Does the introduction provide sufficient background and include all relevant references? |
Must be improved |
Thanks, we have revised the manuscript to add references and more background information |
||
|
Are all the cited references relevant to the research? |
Must be improved |
Thanks, we have added references in our revised manuscript |
||
|
Is the research design appropriate? |
Can be improved |
Thanks, we have explained the research design in the methods section in the revised manuscript. |
||
|
Are the methods adequately described? |
Can be improved |
We have revised our manuscript to clarify the methods |
||
|
Are the results clearly presented? |
Yes |
Thanks! |
||
|
Are the conclusions supported by the results?
|
Yes |
Thanks!. |
||
|
3. Point-by-point response to Comments and Suggestions for Authors |
|
|||
|
Comment 1: Panels do not have scale bars. This element is crucial when analyzing processes affecting clone and tissue size. |
|
|||
|
Response 1: Thank you for pointing this out. We agree with this comment. Therefore, we have added scale bars to all figures in the revised manuscript.
|
|
|||
|
Comment 2: Fig 1C. What part of the disc is that? The authors should have used a general marker as DAPI or phalloidin to show the shape of the disc. |
|
|||
|
Response 2: Thanks for your comment. Fig. 1 C is an image of the eye disc bearing RasV12, scrib2- clones at 40X magnification. To clarify, we have added a lower magnification image in Supplementary Figure 1 (Fig. S1B-B”).
Comment 3: What is shown in 1E? I only see some green patches but the shape of the disc can´t be observed. Response 3: Thanks for pointing this out, we have marked the outline of the wing disc (yellow dashed line). Comment 4: ”however the discs remain monolayered and clones did not form tumours (Fig. S1B)” where is it shown that the clones are monolayered? I can´t find any panel showing that. Response 4: Thanks, we have added this information in the revised manuscript in Fog. S1 B-B” to show the YZ sections for the eye disc.
Comment 5: “In comparison to wild-type clones (Fig. 1B-B’), in RasV12,scrib- clones a bulk of cell death was induced in the wild-type cells surrounding the clone (Fig. 1C, C’).” This is partially true. There is some clear caspase activity in GFP positive cells close to the arrowhead at the bottom. Response 5: Thanks, we have revised the manuscript to indicate that the Caspase activity is seen both inside and outside the clone, however, many more cells outside or at the border of the clones are dying.
Comment 6: “Similarly in the wing discs, compared to wild-type (Fig.1D) RasV12,scrib- 187 clones grew to large invasive tumours (Fig. 1E).” I can´t see any invasion in that panel. Response 6: Agreed. We have added a new figure (Fig. S1D) to depict the large invasive clones in the wing discs. This was also shown by Atkins et al., 2016 (doi: 10.1016/j.cub.2016.06.035)
Comment 7: The authors refer recurrently to Ay-Gal4. Is that correct or do they mean eyeless-gal4 (ey-Gal4). Response 7: Sorry about the confusion. AyGal4 refers to the ‘Flp-out’ Gal4 construct Act>y+>Gal4 published in Ito et al., 1997, which is also referred to as AyGal4.
Comment 8: “Increased Yki activity due to Yki overexpression in ‘flp-out’ (AyGAL4>Yki) clones showed robust induction of diap1-lacZ (Fig. S1D).” It´s impossible to see Diap1 expression in FS1. The GFP levels are masking the red signal. Response 8: Thanks for the suggestion. We have presented the split channel to show induction of diap1-lacZ in Fig S1F, F’.
Comment 9: The authors claim tha Dronc is induced in Ras-scrib clones, F1G. But Dronc seems to be upregulated in a patch of cells with a similar shape of that clone in wild type discs (F1F&F´). Response 9: We have quantified the expression of Dronc in the clones (GFP positive) and normalized it with the GFP-negative wild-type expression of Dronc to assess changes in Dronc expression. The dot plots represent the normalized quantification.
Comment 10: Wg is claimed to be induced in those clones. However, it is upregulated in some cells of the clone (Fig1J&J´). How do the authors reconcile that with their model? The same coment applies for Dronc-lacZ (F1M). Response 10: In wild-type eye discs, Wg expression is limited to a small group of cells on the dorsal and ventral margin of the eye disc, thus any induction in the eye disc outside of this endogenous domain is significant. The induction of Wg is not uniform in the clone – which may reflect the heterogeneity of tumor cells and differences in their progressive states. This is why several other markers are not uniformly and cell autonomously induced in RasV12 scrib- clones, for example, JNK and its target gene MMP1 (Uhlirova and Bohmann, 2006; Cordero et al., 2010; Srivastava et al., 2007; Leong et al., 2009). This may also explain why dronc-lacZ (Fig. 1 M) is not uniformly induced in the tumor clones.
|
|
|||
|
4. Response to Comments on the Quality of English Language |
|
|||
|
Point 1: It can be improved |
|
|||
|
Response 1: The manuscript has been edited by a native English speaker ( all edits are marked in red in the revised manuscript).
|
|
|||
|
5. Additional clarifications |
|
|||
|
|
|
|||
Round 2
Reviewer 1 Report
Comments and Suggestions for Authors
I am satisfied with the response of the authors except in two points (copied below in blue, with the authors' response), where I could not find in the new manuscript the changes described by the authors. Could they please point more specifically at where those changes were made? For instance, I could not find the reference referred to as "ref 30" below (new reference #38, I think) cited anywhere in the discussion, but I may be missing it systematically.
Minor points:
i. The fact that caspases have non-apoptotic roles is well known and is starting to be characterised in detail (see, for instance, the work of LA Baena-López), so it is a bit unnecessary to repeat over and over that your finding of Dronc being upregulated in growing tumours is 'paradoxical'. Rather, the work of the people who have shown that this is the not the case should be cited.
Response: Thanks, we have added citations to include the apoptotic and nonapoptotic roles of Dronc in the context of cancer, and toned down the ‘paradoxical role’.
xiv. In the discussion, you claim to "have found that JNK, Yki, Dronc, and Wg were all required for aggressive growth of RasV12,scrib- induced tumours" (lines 524-5). However the role of JNK and Hippo pathways in these clones had been described a while ago in Igaki et al., 2006 (10.1016/j.cub.2006.04.042) and in Doggett et al., 2011 (10.1186/1471-213X-11-11). [Those are the references I could find. There may be others.] Igaki et al in particular is cited in the manuscript as ref 30 but it is not cited in the context of this finding. This should be corrected.
Response: Thanks for your comment. We have added these and other references in response to your comments.
Author Response
For research article
Manuscript ID: cancers-2734242
Title: A Tumour-Specific Molecular Network Promotes Tumour Growth in Drosophila by Enforcing a JNK-YKI Feedforward Loop
|
Response to Reviewer 1 Comments
|
|
|||
|
1. Summary |
|
|
|
|
|
Thank you very much for taking the time to review this manuscript. Please find the detailed responses below and the corresponding revisions in red in the re-submitted files.
|
|
|||
|
2. Questions for General Evaluation |
Reviewer’s Evaluation |
Response and Revisions |
|
|
|
Does the introduction provide sufficient background and include all relevant references? |
Can be improved |
We have added citations. |
|
|
|
Are all the cited references relevant to the research? |
Yes |
Thank you |
|
|
|
Is the research design appropriate? |
Yes |
Thank you |
|
|
|
Are the methods adequately described? |
Yes |
Thank you |
|
|
|
Are the results clearly presented? |
Yes |
Thank you |
|
|
|
Are the conclusions supported by the results?
|
Yes |
Thank you |
|
|
|
3. Point-by-point response to Comments and Suggestions for Authors |
|
|||
|
Comments 1: I am satisfied with the response of the authors except in two points (copied below in blue, with the authors' response), where I could not find in the new manuscript the changes described by the authors. Could they please point more specifically at where those changes were made? For instance, I could not find the reference referred to as "ref 30" below (new reference #38, I think) cited anywhere in the discussion, but I may be missing it systematically. Minor points: i. The fact that caspases have non-apoptotic roles is well known and is starting to be characterised in detail (see, for instance, the work of LA Baena-López), so it is a bit unnecessary to repeat over and over that your finding of Dronc being upregulated in growing tumours is 'paradoxical'. Rather, the work of the people who have shown that this is the not the case should be cited. xiv. In the discussion, you claim to "have found that JNK, Yki, Dronc, and Wg were all required for aggressive growth of RasV12,scrib- induced tumours" (lines 524-5). However the role of JNK and Hippo pathways in these clones had been described a while ago in Igaki et al., 2006 (10.1016/j.cub.2006.04.042) and in Doggett et al., 2011 (10.1186/1471-213X-11-11). [Those are the references I could find. There may be others.] Igaki et al in particular is cited in the manuscript as ref 30 but it is not cited in the context of this finding. This should be corrected. |
|
|||
|
|
Response 1: Thank you for your comment. The references are included in the revised manuscript as follows: (a) Igaki et al (2006) is Reference 38 in the manuscript. 38. Igaki T, Pagliarini RA, Xu T. Loss of Cell Polarity Drives Tumor Growth and Invasion through JNK Activation in Drosophila. Current Biology. 2006;16: 1139–1146. doi:10.1016/j.cub.2006.04.042 · Is cited in the Results, lines 183-185 o “Compared to wild-type (Fig. 1A), scrib- clones grew poorly (Fig. S1A) [38]. Given that scrib- clones are competed out due to cell competition mediated apoptosis [12,38],---" · Is cited in the Results, lines 206-208 o “In contrast, elimination of scrib- cells by cell competition involved JNK-dependent suppression of Yki activity [12,38,46–48].” · Is cited in the Results, lines 390-391 o “Loss of polarity (scrib) alone is insufficient to induce aggressive growth in somatic clones (Fig. 1, S1) [8,9,12,38], therefore,---" · Is cited in the Results, lines 441-442 o “Expression of Wg was robustly induced in the scrib- clones (Fig. 6C, C’) whereas pJNK levels appear upregulated in both scrib- and RasV12 clones (Fig. 6D, D’)[10,38].” |
|||
|
|
· Is cited in the Discussion, lines 492-493 o “Cooperative interactions between oncogenic Ras and loss of scrib- (RasV12,scrib-) have been elegantly modelled using in-vivo mosaic tumour models in Drosophila [8,9,12,38], and---" · Is cited in the Discussion, lines 511-512 o “JNK is a pivotal stress-responsive kinase that promotes malignant transformation and metastasis of tumours [38,45].” · Is cited in the Discussion, line 540-541 o “Previous studies have shown a role for the JNK and Hippo pathway in RasV12 scrib- clones [38,41].”
|
|||
|
|
(b) Doggett et al., 2011 is Reference 45. 45. Doggett K, Grusche FA, Richardson HE, Brumby AM. Loss of the Drosophila cell polarity regulator Scribbled promotes epithelial tissue overgrowth and cooperation with oncogenic Ras-Raf through impaired Hippo pathway signaling. BMC Dev Biol. 2011;11: 57. doi:10.1186/1471-213X-11-57 |
|||
|
· Is cited in the Results, lines 205-206 o “JNK and Yki are known to interact in a context-dependent manner [12–14,41], and increased JNK or Yki activity has been linked to tumour growth [12,42–45].” |
|
|||
|
|
· Is cited in the Discussion, lines 511-512 o “JNK is a pivotal stress-responsive kinase that promotes malignant transformation and metastasis of tumours [38,45].”
(c) In the context of non-apoptotic roles of caspases, Dr. Baena-Lopez’s work has been cited as follows: Reference 40 reviews cell competition and known regulators including caspases. 40. Vincent J-P, Fletcher AG, Baena-Lopez LAl. Mechanisms and mechanics of cell competition in epithelia. Nat Rev Mol Cell Biol. 2013;14: 581–591. doi:10.1038/nrm3639 · Is cited in the Results, lines 201-204 o “Therefore, as a next step we tested if JNK, Yki, Dronc and Wg, markers that have previously been linked to cell survival and proliferation during cell competition [22–24,40], were also affected in the RasV12 scrib- tumours.”
|
|||
Reference 79, reports non-apoptotic role of Dronc (Caspase2/9) and its interaction with JNK signaling in the tumors generated by coactivation of EGFR and JAK/STAT pathways.
- Xu DC, Wang L, Yamada KM, Baena-Lopez LA. Non-apoptotic activation of Drosophila caspase-2/9 modulates JNK signaling, the tumor microenvironment, and growth of wound-like tumors. Cell Reports. 2022;39: 110718. doi:10.1016/j.celrep.2022.110718
- Is cited in the Discussion, lines 535-537
- In addition, misregulation of the Hippo, JNK or Wg pathway is also linked to activation of caspases mediated apoptosis, and recently mild caspase induction was shown to promote tumour growth [77–79].